# The VM2D Open Source Code for Two-Dimensional Incompressible Flow Simulation by Using Fully Lagrangian Vortex Particle Methods

Ilia Marchevsky [1,2,*,†] , Kseniia Sokol [1,†] , Evgeniya Ryatina [1,†] and Yulia Izmailova [1]

1   Department of Applied Mathematics, Bauman Moscow State Technical University, Moscow 105005, Russia
2   N. I. Lobachevskii Institute of Mathematics and Mechanics, Kazan Federal University, Kazan 420008, Russia
*   Correspondence: iliamarchevsky@mail.ru
†   These authors contributed equally to this work.

**Abstract:** This article describes the open-source C++ code VM2D for the simulation of two-dimensional viscous incompressible flows and solving fluid-structure interaction problems. The code is based on the Viscous Vortex Domains (VVD) method developed by Prof. G. Ya. Dynnikova, where the viscosity influence is taken into account by introducing the diffusive velocity. The original VVD method was supplemented by the author's algorithms for boundary condition satisfaction, which made it possible to increase the accuracy of flow simulation near the airfoil's surface line and reduce oscillations when calculating hydrodynamic loads. This paper is aimed primarily at assessing the efficiency of the parallelization of the algorithm. OpenMP, MPI, and Nvidia CUDA parallel programming technologies are used in VM2D, which allow performing simulations on computer systems of various architectures, including those equipped with graphics accelerators. Since the VVD method belongs to the particle methods, the efficiency of parallelization with the usage of graphics accelerators turns out to be quite high. It is shown that in a real simulation, one graphics card can replace about 80 nodes, each of which is equipped with 28 CPU cores. The source code of VM2D is available on GitHub under GNU GPL license.

**Keywords:** fluid-structure interaction problem; incompressible flow; Lagrangian method; Viscous Vortex Domain method; MPI; OpenMP; Nvidia CUDA; open-source code; VM2D

**MSC:** 76D17; 76M23; 76-04; 65Y05



## 1. Introduction

Lagrangian vortex particle methods (VPM) [1–3] are a powerful tool for numerical simulation in number of applications: from flow modeling for animations (e.g., 3D smokes [4]) to industrial aerodynamics problems solving, including averaged and unsteady hydrodynamical loads estimation for various structures, vortex wakes simulation, hydroelastic oscillations simulation of structural elements which interact with the flow—Fluid-Structure Interaction problems (FSI). The range of the problems where traditional modifications of vortex methods are applicable is limited by the flow regimes with low subsonic Mach numbers, when the influence of flow compressibility can be neglected.

For such problems, especially FSI ones, vortex methods can be very efficient, at least in comparison to "traditional" mesh-based methods, where it is necessary to deform or reconstruct mesh at every time step due to body motion. Moreover, for vortex methods, computational cost for FSI problems remains nearly the same as for flow simulation around immovable body. Note that an accurate iterative coupling strategy has been suggested in [5], based on the added masses computation, which is extremely easy for implementation and provides a numerically stable simulation with large time steps for bodies of arbitrary low mass.

Reviewing briefly advantages of VPM, the following features can be pointed out:

- External and unbounded flows can be simulated with exact satisfaction of the perturbations decay boundary condition at infinity;
- FSI problems can be considered with arbitrary translations, rotations, and deformations of the streamlined surface;
- The most part of computational resources is "concentrated" in the part of the flow domain with non-zero vorticity; such area is usually rather compact;
- VPM belongs to a class of particle-based methods, where Lagrangian particles are considered as vorticity carriers; it follows therefore that VPM provides rather small numerical diffusion.

Although VPM has a long history [6–8] dating back to the 1930s, starting from the well-known work of Rosenhead [9], nowadays, there are just a few known software implementations available to a wide range of researchers, which are based on modern mathematical models and computing technologies. The most part of available codes that implement various modifications of VPM appeared only in recent years.

The vvflow [10] code that became available in 2018 implements the Viscous Vortex Domains method [11] for 2D flow simulations. It allows to solve FSI problems for systems of non-deformable airfoils with elastic constraints. This code is open-source now; however, the only OpenMP technology is implemented for parallelization that allows computations only on shared memory multicore systems.

In 2018, open source codes Omega2D [12] and Omega3D [13] appeared, which implement VPM for plane and spatial flows simulation [14]. These codes are positioned by authors as a platform for the development and implementation of various VPM modifications; much attention is given to visualization issues. At the moment, these codes do not have the ability to simulate the flow around moving airfoils/bodies and the issue of the accuracy of calculating the hydrodynamic loads acting on the streamlined surfaces is not entirely clear (the ability to calculate them is generally disabled by default). In addition, when calculating the mutual influence of vortex particles, only the Biot–Savart law is used (i.e., algorithm of quadratic computational complexity), computational speedup can be achieved only by connecting external libraries that allow part of the computational work to be performed on graphical accelerators.

In 2019, the FLOWVPM code appeared, which is part of the larger FLOWUnsteady project [15], freely available with source code written in Julia. The package allows for simulation of spatial flows and flows around bodies (essentially three-dimensional), including moving ones with complex geometric shapes; the description of the models and methods implemented in it is given in [16]. Technologies for parallelization of the computations are not used in the available version of the code; the Fast Multipole Method (FMM) is used for the calculation of the velocities of vortex particles. Vortex blobs are used as vortex particles; thus, the vorticity field is generally non-solenoidal.

It is also known about VXflow—the code developed by Prof. G. Morgenthal [17], which is actively used by his scientific group (however, it is not freely available). It allows for solving problems of industrial aerodynamics of buildings and structures in a quasi-3D setting (by using the flat cross-section method). This code implements the random walk method [18] and the original modification of the fast method based on Fast Fourier Transform (FFT) for calculating the velocities of vortex particles [19]; the basic models and algorithms underlying the code are described in [20]. Algorithms for parallelization of the computations on cluster systems are not implemented in VXflow, but the usage of OpenCL technology allows for performing computations on graphics cards.

Thus, freely available codes still have very modest capabilities, both in terms of the implemented mathematical models and in the sense of ensuring high computing performance, both at the algorithmic level and due to the capabilities of modern multiprocessor computers, as well as graphics accelerators. So, it is assumed that in order to create an efficient computational tool based on VPM, it is necessary to provide the possibility of using modern and the most efficient mathematical models and numerical algorithms, to-

gether with the computational capabilities of modern multiprocessor computers of various architectures. As an attempt to create such a tool, the code VM2D [21] is developed, which allows simulating two-dimensional incompressible flows around movable and immovable airfoils and solving coupled FSI problems. The source code is open and freely available on GitHub: http://github.com/vortexmethods/VM2D (accessed on 22 February 2023).

A brief comparison of the mentioned vortex particle codes is shown in Table 1.

**Table 1.** Brief comparison of the known codes.

| Code Name | vvflow | Omega2D | VXflow | VM2D |
|---|---|---|---|---|
| Open-source | 2018 | 2018 | — | 2017 |
| Problems | FSI problems for systems of non-deformable movable airfoils | immovable airfoils, visualization | 2D and quasi-3D problems of industrial aerodyn. of buildings | FSI problems for systems of non-deformable movable airfoils |
| Viscous term approximation | viscous vortex domains method (based on diffusive velocity method) | adaptive vorticity redistribution method (VRM) | random walk method | viscous vortex domains method (based on diffusive velocity method) |
| Velocity computation method | Barnes–Hut method $O(N \log N)$ | Biot–Savart method $O(N^2)$ | FFT-based method $O(N \log N)$ | Biot–Savart method $O(N^2)$ |
| Parallelization | CPU OpenMP | CPU OpenMP | CPU+GPU OpenCL | CPU+GPU OpenMP+MPI+CUDA |

The advantages of the developed VM2D code are connected with implementation of improved higher-order schemes for vorticity generation simulation, consideration of wide range of 2D problems, including FSI problems, a convenient interface for multiple tasks solution on cluster systems. The code is parallelized with OpenMP, MPI and CUDA technologies (including their mutual usage). However, the implemented algorithms are direct, i.e., provide $O(N^2)$ computational complexity. Fast methods for different parts of the algorithm based on the Barnes–Hut and FMM methods, as well as their "hybrid" modification, will be implemented in future versions; now, they are implemented just partially, only for velocity computation in OpenMP mode.

This paper aims to describe the VM2D code, modifications of VPM, numerical schemes, and algorithms that underlie it as well as to give an idea of the range of its applicability and computational efficiency.

The paper is organized as follows. Section 2 describes the problem of modeling two-dimensional flows and solving FSI problems. The governing equation is the Navier–Stokes equation, which can be transformed into the vorticity transfer equation. The boundary condition at the airfoil's surface line is obtained from the no-slip condition and is written as an integral equation with respect to the vortex sheet intensity. Section 3 gives a brief history of the development of VPM, a description of their modification—the Viscous Vortex Domain (VVD) method, numerical schemes for solving the boundary integral equation, and for simulating vorticity transfer in the flow domain. The solutions of model problems are also presented in Section 3; the efficiency of the described approaches is estimated for them. Section 4 describes the structure of the VM2D code and the input/output files, and also shows the efficiency of parallelization using the CPU and GPU.

## 2. The Governing Equations

In the present paper, we consider only two-dimensional flows; so, hereinafter we use the term "airfoil" instead of "body". Having in mind that the range of applicability of 2D simulations is rather narrow (nevertheless, they are rather popular in engineering practice, including quasi-3D simulations according to the flat cross-section method), we note that despite the significant differences between a plane and spatial flows, the major part of the methods and approaches can be generalized and transferred onto 3D case.

### 2.1. The Navier-Stokes Equations

The VPM are based on consideration the vorticity $\boldsymbol{\Omega} = \text{curl } \boldsymbol{V}$ as a primary computed variable, so the Navier–Stokes equations [22]

$$\nabla \cdot \boldsymbol{V} = 0, \quad \frac{\partial \boldsymbol{V}}{\partial t} + (\boldsymbol{V} \cdot \nabla)\boldsymbol{V} = -\frac{\nabla p}{\rho} + \nu \Delta \boldsymbol{V}, \tag{1}$$

where $\boldsymbol{V}$ and $p$ are velocity and pressure fields; $\rho$ and $\nu$ are flow density and kinematic viscosity that are assumed to be constant; $\Delta$ is the Laplacian operator, can be written down in the following form after applying the curl operator:

$$\nabla \cdot \boldsymbol{V} = 0, \quad \frac{\partial \boldsymbol{\Omega}}{\partial t} + \nabla \times (\boldsymbol{\Omega} \times \boldsymbol{V}) = \nu \Delta \boldsymbol{\Omega}.$$

Note that in 2D flows, the vorticity vector $\boldsymbol{\Omega}$ is always orthogonal to the flow plane, so one can write down $\boldsymbol{\Omega} = \Omega \boldsymbol{k}$, where $\boldsymbol{k}$ is the corresponding unit vector, and thus, the vorticity can be considered as a scalar variable. This property allows for the following representation of the viscous term:

$$\nu \Delta \boldsymbol{\Omega} = \nabla \times (\boldsymbol{W} \times \boldsymbol{\Omega}), \quad \boldsymbol{W} = \nu \frac{(\nabla \times \boldsymbol{\Omega}) \times \boldsymbol{\Omega}}{|\boldsymbol{\Omega}|^2} = -\nu \frac{\nabla \Omega}{\Omega},$$

where $\boldsymbol{W}$ is the so-called diffusive velocity field [11,23–26].

As a result, the equation that describes the vorticity evolution has the form

$$\frac{\partial \boldsymbol{\Omega}}{\partial t} = \nabla \times \big((\boldsymbol{V} + \boldsymbol{W}) \times \boldsymbol{\Omega}\big), \tag{2}$$

or, the same for the "scalar" vorticity,

$$\frac{\partial \Omega}{\partial t} + \big((\boldsymbol{V} + \boldsymbol{W}) \cdot \nabla\big)\Omega = 0,$$

and can be considered as a vorticity transfer equation in the flow domain: existing vorticity is transferred with the velocity field $(\boldsymbol{V} + \boldsymbol{W})$ without changing its intensity. This means that there are no sources of vorticity generation in the flow domain. In order to distinguish clearly two velocity fields, we hereinafter call the field $\boldsymbol{V}$ as "convective" velocity.

Note that application of the curl operation allows one to avoid considering the pressure field, however, in most applications, it is necessary to reconstruct it or at least to compute pressure-based forces and torque acting on the airfoil. It can be easily done by using the Cauchy–Lagrange integral analogue [27] or some formulae for integral loads, derived in [5,28–30] for a wide range of problem types.

### 2.2. Vorticity and Velocity Fields

While for a given velocity field the vorticity field can be obtained directly as a spatial derivative (curl), the inverse operator, i.e., the possibility of the velocity field reconstruction for a given vorticity field is less trivial. It is based on the so-called Generalized Helmholtz Decomposition (GHD) [31–33], which can be considered also as a generalization of the well-known Biot–Savart law:

$$\alpha(\boldsymbol{\rho})\boldsymbol{V}(\boldsymbol{\rho}) = \boldsymbol{V}_\infty + \int_F \big(\boldsymbol{\Omega}(\boldsymbol{\xi}) \times \boldsymbol{Q}(\boldsymbol{\rho} - \boldsymbol{\xi})\big)dS_\xi + \int_F D(\boldsymbol{\xi})\,\boldsymbol{Q}(\boldsymbol{\rho} - \boldsymbol{\xi})\,dS_\xi +$$

$$+ \oint_K \Big((\boldsymbol{n}(\boldsymbol{\xi}) \times \boldsymbol{U}_K(\boldsymbol{\xi})) \times \boldsymbol{Q}(\boldsymbol{\rho} - \boldsymbol{\xi})\Big)dl_\xi + \oint_K (\boldsymbol{n}(\boldsymbol{\xi}) \cdot \boldsymbol{U}_K(\boldsymbol{\xi}))\,\boldsymbol{Q}(\boldsymbol{\rho} - \boldsymbol{\xi})\,dl_\xi. \tag{3}$$

Here, $\alpha(\boldsymbol{\rho})$ is a parameter that is equal to 1 in the flow domain and equal to 0 inside the airfoils (note, that the presented integral representation of velocity field remains in force

inside the airfoils, where there is no flow formally); at the airfoil boundary $\alpha(r) = \frac{\chi(r)}{2\pi}$, $r \in K$, where $\chi$ is the measure of the angle between left and right tangent vectors at the point $r$, i.e., on smooth parts of the airfoil's surface line $\alpha(r) = 1/2$. Vector $V_\infty$ denotes the incident flow velocity that presents in external flows and is absent in internal flows; $U_K$ is the airfoil (wall) velocity; $K$ is the airfoil's surface line; $F$ is the flow domain.

Vector field $\Omega(\rho) = \nabla \times V(\rho)$ and scalar field $D(\xi) = \nabla \cdot V(\rho)$ mean the curl and divergence of the velocity field, respectively. Vector $n(\xi)$ denotes the outer (with respect to the airfoil) normal unit vector at the corresponding point; at corner and cusp points two equations that follow from (3) should be considered: left and right limits in proximity to the point $r$. The kernel in the integrals $Q(\rho - \xi)$ means the gradient of the principal solution of the Laplacian equation in 2D space, taken with the opposite sign:

$$Q(\rho) = -\nabla G(\rho) = \frac{1}{2\pi} \frac{\rho}{|\rho|^2}.$$

The contour integrals in (3) are calculated over the airfoils' surface lines; the limit value of the flow velocity on them is required to be equal to the boundary velocity $U_K$, according to the no-slip boundary condition:

$$V(r) = U_K(r), \quad r \in K. \tag{4}$$

The perturbation decay condition considered at infinity,

$$V(r) \to V_\infty, \quad |r| \to \infty, \tag{5}$$

is satisfied automatically for (3), and there is no need to bound the computational domain somehow for external flow simulations. This feature can be considered as one of the advantages of the VPM.

Taking into account that the flow is incompressible, i.e., $D(\xi) \equiv 0$, the corresponding integral in (3) vanishes, while the resulting field $V(\rho)$ expressed by (3) remains divergence-free for arbitrary vorticity distribution in the flow domain $\Omega(\rho)$, $\rho \in F$, and arbitrary motion of the airfoil boundaries, described by velocity $U_K(r)$, $r \in K$. It is noted in [31] that the kinematic consistency condition should be satisfied between $\Omega(\rho)$ field and surface velocity $U_K(r)$: vorticity distribution cannot be chosen arbitrary because its influence, expressed by the corresponding term in (3), provides at the airfoil's surface line exactly the same velocity $V(r) = U_K(r)$, $r \in K$.

### 2.3. Vorticity Generation

Summarizing the above-expressed relations, one can conclude that the Navier–Stokes Equations (1) in the flow domain are satisfied by vorticity transfer with summary convective and diffusive velocity, where convective velocity is expressed according to (3), and the perturbation decay boundary condition (5) is satisfied automatically.

Thus, it is necessary to satisfy the no-slip boundary condition (4). It is well-known that it is impossible to re-formulate correctly the no-slip condition in terms of vorticity [34,35] (this issue can be essential in attempts of flow simulation, where the vorticity and stream function are considered as primary variables), so we leave the condition (4) "as is", substituting there the expression for the convective velocity that follows from (3). The no-slip condition satisfaction is provided by vorticity generation at the airfoil's surface line, which makes it possible to introduce the so-called vorticity flux $\psi(r)$, $r \in K$, equal to the vorticity generation rate on the surface line. One can write down the integral equation for $\psi(r)$ [5,30]; however, such approach does not seem to be useful in practical computations.

Instead, we take into account the estimations, derived in [36,37], according to which the rate of diffusion of the vorticity, being generated at the airfoil's boundary, into the flow domain due to viscosity effect, is rather low that makes it possible to assume that the most part of vorticity, being generated during a small time period (one time step in the

computational algorithm), remains contained in the thin near-wall layer. Neglecting the thickness of this layer, we deal with a thin vortex sheet of unknown intensity $\gamma(r) = \gamma(r)k$ which is considered as part of vorticity in the flow domain, which therefore consists of distributed part $\Omega(\rho)$, $\rho \in F$, and concentrated part $\gamma(r)$, $r \in K$. Now, $\Omega(\rho)$ means the result of the solution of Equation (2) at the end of the time step, i.e., the result of the transfer of the vorticity that had already existed at the beginning of the time step.

Introducing such representation in (3) and taking into account that the velocity is calculated now at the airfoil's surface line, one can notice that all the contour integrals now become singular, so the Cauchy principal values should be considered [38], which we denote with a traditional integral sign. Moreover, the vortex sheet $\gamma(r)$ provides jump discontinuity for the tangent velocity at the surface line, and since we consider the vortex sheet as a part of vorticity distribution in the flow, the limit value of the corresponding singular integral expressed through the Cauchy principal value according to the Sochocki–Plemelj theorem [39,40] should be calculated as

$$V^{\gamma}_{-}(r) = \oint_{K} \frac{\gamma(\xi) \times (r - \xi)}{2\pi|r - \xi|^2} dl_{\xi} - \alpha(r)\gamma(r) \times n(r)$$

Finally, the resulting boundary integral equation (BIE) that expresses the no-slip boundary condition (4) takes the form

$$\oint_{K} (\gamma(\xi) \times Q(r - \xi)) dS_{\xi} - \alpha(r)(\gamma(r) \times n(r)) = f(r), \quad r \in K; \tag{6}$$

the right-hand side is the following:

$$f(r) = \alpha(r)U_K(r) - V_{\infty} - \int_{F} (\Omega(\xi) \times Q(r - \xi)) dS_{\xi} -$$
$$- \oint_{K} (\gamma^{att}(\xi) \times Q(r - \xi)) dl_{\xi} - \oint_{K} q^{att}(\xi) Q(r - \xi) dl_{\xi}, \tag{7}$$

where notations

$$\gamma^{att}(r) = n(r) \times U_K(r), \qquad q^{att}(r) = n(r) \cdot U_K(r),$$

are introduced, since the effect of the corresponding contour integral terms is exactly the same as the influence induced by vortex sheet with intensity $\gamma^{att}(r)$ and source sheet with intensity $q^{att}(r)$. These sheets are called "attached" in order to highlight their non-hydrodynamic nature: their intensities are not caused by the flow, but are fully determined by the airfoil's surface line motion. In order to distinguish clearly the vortex sheets, the above-introduced intensity $\gamma(r)$ we call hereafter "free vortex sheet".

If several airfoils are considered, i.e., the flow domain is bounded by several closed surface lines $K_s$, $s = 1, \ldots, N_S$, the integral equation similar to (6) should be satisfied for each contour, where now the integral $\oint_{K} (\ldots) dl_{\xi}$ means the sum of integrals $\sum_{s} \oint_{K_s} (\ldots) dl_{\xi}$, only one from which is singular.

In general case, the integral Equation (6) has infinite set of solutions [40,41]; in order to pick out the unique one, which has a physical sense, the additional integral condition should be considered together with (6):

$$\oint_{K} \gamma(\xi) dl_{\xi} = \Delta\Gamma_K, \tag{8}$$

where $\Delta\Gamma_K$ means the difference between circulation values for the $U_K$ velocity

$$\Gamma_K = \oint_{K} U_K(\xi) \cdot dl_{\xi},$$

at the ending and beginning of the considered time step. For translational motion of a rigid airfoil $\Delta\Gamma_K = 0$, for rotational motion $\Delta\Gamma_K$ is equal to doubled change of the angular velocity during the time step multiplied by the airfoil area.

As for the integral equation, for multiple airfoils additional conditions similar to (8) should be written down for each of them.

*2.4. Fluid-Structure Interaction Problems*

In case of immovable airfoil or when its law of motion is known, the only unknown variable in (6) is the free vortex sheet intensity $\gamma(\boldsymbol{r})$, while in FSI problems, the airfoil's surface line velocity $\boldsymbol{U}_K(\boldsymbol{r})$ in (7) is also unknown, as well as above introduced attached vortex and source sheet intensities $\gamma^{att}(\boldsymbol{r})$ and $q^{att}(\boldsymbol{r})$. So, in order to solve FSI problems, some splitting scheme implementation is required. The simplest approach—weak coupling strategy—is suitable when the inertial properties of a moving airfoil (mass, moment of inertia) are higher than the added mass and added moment of inertia; for bluff bodies, they can be roughly estimated as the mass and moment of inertia of the displaced fluid. According to the weak coupling strategy, each time step is split into two sub-steps. During the first of them, the velocity of the airfoil's surface line is assumed to be constant, so in boundary integral Equation (6), only the free vortex sheet intensity is unknown; at the second sub-step, dynamical equations for the structure are solved under hydrodynamic loads predetermined from the first sub-step, and $\boldsymbol{U}_K(\boldsymbol{r})$ distribution is updated. Note that the time steps in hydrodynamical and mechanical sub-problems can differ, which means that one of them in general corresponds to several of the others.

For rather light airfoils, the described weak coupling strategy requires simulation with a small time step, however, if the airfoil's inertial properties are smaller than the added mass and added moment of inertia, it anyway leads to numerical instability. In this case, an extremely accurate monolithic strategy [42] can be applied, but it is not easy to implement it in the framework of "universal" code, developed for a rather wide range of problems. So, the semi-implicit coupling strategy based on expressions of hydrodynamic loads derived in [5], which can be called "strongly coupling", seems to be the most suitable. This scheme is iterative (opposite to the monolithic one), however, it is compensated by the simplicity of its implementation and in most cases, 1–2 iterations are enough.

In order to implement the last strongly coupling strategy, the added masses tensor of the airfoil should be known, which components can be rather easily found by solving three (in the most general case) auxiliary problems for an impulsive start of the considered airfoil in horizontal and vertical directions in still media with unit velocity $|\boldsymbol{U}_K| = 1$, and its impulsive start in rotational motion with unit angular velocity $\omega = 1$. It requires solving the integral Equation (6) with the right-hand side (7) where $\boldsymbol{V}_\infty = \boldsymbol{0}$ and $\boldsymbol{\Omega}(\boldsymbol{\rho}) = \boldsymbol{0}$, and all the other boundaries, if they present in the problem, are considered to be immovable at their current positions, so the contour integrals are calculated only for the considered airfoil. In the three described problems the left-hand side remains the same, the right-hand sides are known, and the additional conditions take the form (8), where

$$\Delta\Gamma_K = -\oint_K \gamma^{att}(\boldsymbol{\xi})\,dl_{\xi},$$

which means that $\Delta\Gamma_K$ is non-zero only for the "rotational" case, and it is equal to doubled airfoil area with an opposite sign (since the angular velocity is unity). As a result, the components of the added mass tensor for two-dimensional flow are the following [5]:

$$\lambda_{dx} = \oint_K \rho(y - y_0)(\gamma_d(\boldsymbol{r}) + \gamma_d^{att}(\boldsymbol{r}))\,dl, \qquad \lambda_{dy} = -\oint_K \rho(x - x_0)(\gamma_d(\boldsymbol{r}) + \gamma_d^{att}(\boldsymbol{r}))\,dl,$$

$$\lambda_{d\omega} = -\frac{1}{2}\oint_K \rho\big((x - x_0)^2 + (y - y_0)^2\big)\big(\gamma_d(\boldsymbol{r}) + \gamma_d^{att}(\boldsymbol{r})\big)\,dl,$$

where $x$ and $y$ are the abscissa and the ordinate of the point $\boldsymbol{r}$ at the airfoil boundary; $x_0$ and $y_0$ are the base point coordinates, which choice is however not essential for added

masses $\lambda_{xx}$, $\lambda_{xy}$ and $\lambda_{yy}$; lower index $d$ has values $x$, $y$ and $\omega$ for the corresponding motion directions; $\gamma_d(\check{\boldsymbol{\xi}})$ and $\gamma_d^{att}(\check{\boldsymbol{\xi}})$ denote free and attached vortex sheets intensities, respectively, that correspond to the airfoil motion in the $d$-th direction.

## 3. Vortex Particle Methods

A review of the current state-of-art for vortex methods can be found in [1,7,43]. In the present paper, we just point to the most famous scientists whose contribution seems to be the most essential and then describe the most significant details of the algorithm that is implemented in the VM2D code.

### 3.1. Brief History of Vortex Methods for 2D Flow Simulation

The vortex methods have a long history, dating back to 1906 when Prof. N. E. Zhukovsky (1847–1921) discovered the fundamental principle, according to which an immovable airfoil influences the inviscid incompressible flow just as attached vortex sheet placed on its surface line [44]. So, it is possible to replace an airfoil with a vortex sheet and to determine somehow its intensity. Prof. L. Rosenhead (1906–1984) was a pioneer, who suggested using this principle in numerical simulations [9,45]. Vortex methods for inviscid flow simulation (both 2D and 3D) have been developed significantly by Prof. V. M. Falkner (1897–1965) [46] and followers, and independently, by Prof. S. M. Belotserkovsky (1920–2000) and co-authors [47]; these modifications are known as "Vortex Lattices Method (VLM)" and "Discrete Vortex Method (DVM)". The mathematical background of such methods, especially connected with the boundary integral equations solution, was significantly developed independently by Profs. J. T. Beale and A. Majda (1949–2021) [48,49] and Prof. I. K. Lifanov [40,47] (1942–2016) with followers.

The concept of vorticity flux on the airfoil's surface in viscous flow was suggested by Prof. M. J. Lighthill [9] (1924–1998) and then developed by many scientists. The random walk stochastic method for vortex particle motion simulation in viscous media was suggested by Prof A. J. Chorin [18] and now seems to be the most popular one. The deterministic diffusive velocity method that was described above (2) dates back to the research of Prof. A. A. Fridman (1888–1925); the idea and current state-of-art of the diffusive velocity approach are discussed in detail in [50], and it was implemented in a numerical algorithm by Profs. Y. Ogami and T. Akamatsu [26]. Prof. G. Ya. Dynnikova suggested the Viscous Vortex Domains (VVD) method, which is based on diffusive velocity introduction and implements the original approach for its computation [23,24]. Hereinafter, we consider the VVD method as basic, while some operations, which are not connected directly to the diffusive velocity reconstruction, are implemented according to original algorithms developed by the authors. Certainly, there are also a lot of brilliant ideas and approaches in vortex methods, developed by many scientists; the most significant are described in the above-mentioned reviews.

### 3.2. Vortex Particle Method Algorithm Based on the VVD Method

In this section, we describe the algorithm of the vortex particle method that is based on the Viscous Vortex Domains method and is implemented in the developed code VM2D.

#### 3.2.1. Vorticity Representation

Vorticity distribution in the flow is simulated by a set of $N$ vortex particles—elementary vorticity carriers, which correspond to circular vortices with some given shape of vorticity distribution, for example, Rankine's vortex or Lamb's vortex (Figure 1). Note that the Rankine's and Lamb's vortices can be considered just as the smoothing technique to avoid unbounded grows of the velocity in proximity to point vortex with singular vorticity distribution $\Omega(\boldsymbol{\rho}) = \Gamma\delta(\boldsymbol{\rho} - \boldsymbol{\rho}_0)$, where $\Gamma$ is total vorticity; $\boldsymbol{\rho}_0$ is position of the point vortex; $\delta$ is the Dirac function. Some other ways of smoothing the velocity profile generated by point vortex can be found in [2]. However, in more complicated models, e.g., for high-Reynolds turbulent flows simulation in the framework of Large Eddy Simulation (LES)

approach [51], such smoothing plays role of the spatial filter size, which is not constant and changes in time.

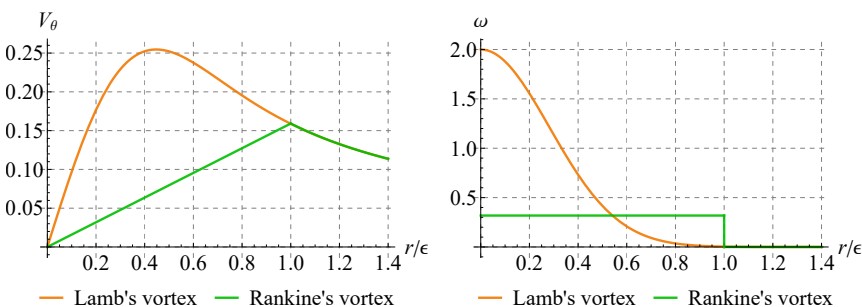

**Figure 1.** Circumferential velocity induced by Rankine's and Lamb's vortices with unit circulation and vorticity distributions; $\epsilon$ is Rankine's vortex radius, Lamb's vortex parameters are such that more, than 0.998 of total vorticity is inside the $\epsilon$-circle; difference of velocities at $r/\epsilon > 1$ is less than 0.2%.

Vorticity in the free vortex sheet is assumed to be represented as some distribution—piecewise-constant or piecewise-linear over rectilinear parts of the discretized airfoil's contour, which we hereinafter call "panels" (Figure 2). Similar piecewise-constant or piecewise-linear representation is considered also for the attached vortex and source sheets.

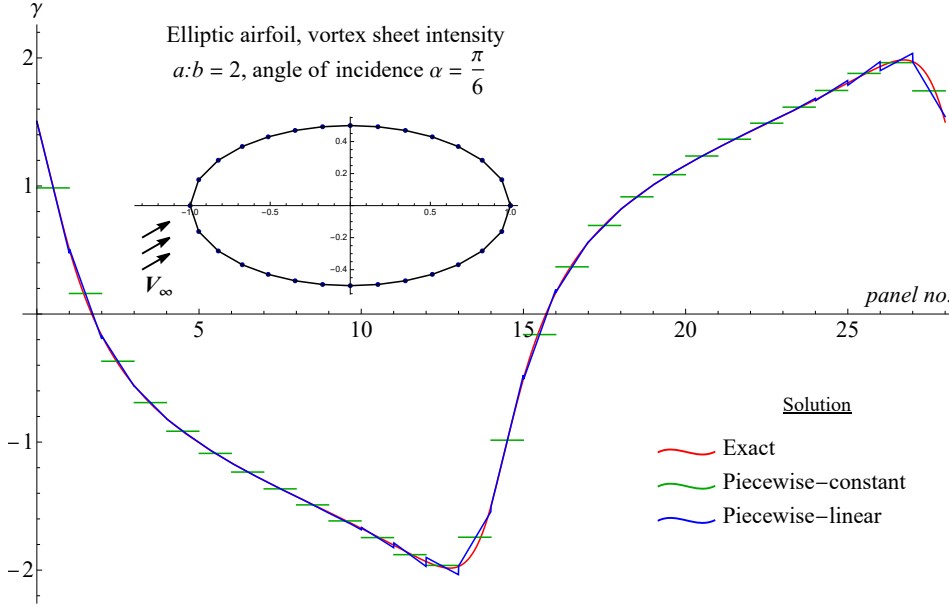

**Figure 2.** Elliptical airfoil with 2:1 semiaxes ratio at angle of incidence $\alpha = \pi/6$; discretization is very coarse, number of panels $N_p = 28$. Free vortex sheet intensity is shown for potential flow: exact distribution (red), piecewise-constant (green), and piecewise-linear (blue) representations.

3.2.2. Convective Velocity Reconstruction

The velocity field, according to (3), consists of several terms, which can be treated as the velocity induced by vorticity distribution in the flow domain $\Omega(\rho)$, by the free and attached vortex sheets and by the attached source sheet, and also the incident flow.

The first part is calculated as follows:

$$V^{\Omega}(\rho) = \int_F \frac{k \times (\rho - \xi)}{2\pi|\rho - \xi|^2}\Omega(\xi)dS_\xi \approx \sum_{w=1}^{N} \frac{k \times (\rho - \rho_w)}{2\pi \max\{|\rho - \rho_w|^2, \epsilon^2\}}\Gamma_w,$$

where $\rho_w$ is position of the $w$-th vortex particle; $\Gamma_w$ is its circulation; smoothing is performed as for the Rankine's vortex.

The contributions of the free and attached vortex sheets and, similarly, the contribution of the attached source sheet are calculated as

$$V^\gamma(\boldsymbol{\rho}) = \oint_K \frac{\boldsymbol{k} \times (\boldsymbol{r} - \boldsymbol{\xi})}{2\pi|\boldsymbol{r} - \boldsymbol{\xi}|^2}\left(\gamma(\boldsymbol{\xi}) + \gamma^{att}(\boldsymbol{\xi})\right)dl_\xi, \quad V^q(\boldsymbol{\rho}) = \oint_K \frac{\boldsymbol{r} - \boldsymbol{\xi}}{2\pi|\boldsymbol{r} - \boldsymbol{\xi}|^2}q^{att}(\boldsymbol{\xi})dl_\xi.$$

The contour integrals are calculated as the sums of integrals over the panels. For piecewise-constant and piecewise-linear integrands $\gamma(\boldsymbol{\xi})$, $\gamma^{att}(\boldsymbol{\xi})$, $q^{att}(\boldsymbol{\xi})$ and rectilinear panels the integrals can be calculated exactly; the necessary formulae are presented in [52–55].

Note that direct integration over vorticity distribution in the flow domain can be too complicated if the number of vortex particles exceeds several tens of hundreds. In order to reduce the computational complexity of the corresponding subroutine, approximate fast methods can be applied. The most popular ones are the following:

- The Barnes–Hut method [56], initially developed for gravitational *N*-body problems, and its modification for vortex methods [57];
- Fast multipole method [58,59], which does not require significant modifications as compared to 2D gravitational problems;
- The fast method, based on the FFT properties [19].

Note that the original modification of the Barnes–Hut method [60], which also can be considered as a hybrid Barnes–Hut/multipole method, is implemented in the `VM2D` code for the problem of convective velocities computation.

### 3.2.3. Vortex Particles Motion

In the framework of the diffusive velocity approach, according to the vorticity evolution Equation (2), the vortex particles should move along the vector lines of the summary convective and diffusive velocity field $V + W$, so the ordinary differential equations system for the positions of vortex particles takes place

$$\frac{d\boldsymbol{\rho}_w}{dt} = \boldsymbol{V}(\boldsymbol{\rho}_w) + \boldsymbol{W}(\boldsymbol{\rho}_w), \quad w = 1, \dots, N, \tag{9}$$

which is solved in the `VM2D` code by using the simplest explicit Euler method. Note that implementation of more accurate methods, e.g., explicit higher-order Runge–Kutta methods, is not trivial due to the necessity of the no-slip boundary condition satisfaction for the convective velocity field not only at the beginning of the time step but also at all its stages. This can be provided by multiple solutions of the BIE (6) with additional condition (8) during one time step. However, for the rather small time steps that are typical for the VPM, the accuracy of Euler method seems to be quite enough.

In order to calculate diffusive velocities $\boldsymbol{W}(\boldsymbol{\rho}_w)$ of the vortex particles, in `VM2D`, the Viscous Vortex Domains method is used, and the formulae presented in [24] with the only difference that the vorticity contained in the free vortex sheet is considered to be distributed over the panels, in opposite to vorticity concentrated in vortex particles.

The transformation of the distributed vorticity from panels into vortex particles in the `VM2D` code is performed according to the following algorithm:

(1) Maximal permitted circulation of the vortex particles $\Gamma_{\max}$ is preliminarily specified;
(2) If the absolute value of total vorticity on the particular panel is higher than $\Gamma_{\max}$, the panel is split into the corresponding number of sub-panels (of different lengths for linear vortex sheet intensity representation) containing equal quantity of vorticity, and the markers are placed at the centers of the sub-panels; in the other case, one marker is placed at the center of the whole panel;
(3) For all the markers, Equations (9) are solved at one time step, as the markers are vortex particles with zero circulation;
(4) Vortex particles are placed at final positions of the markers, their circulations are equal to total vorticity on the sub-panels, corresponding to initial positions of the markers.

In order to compute the diffusive velocities of the vortex particles, according to the VVD method, it is necessary to determine the smoothing scales $\tilde{\epsilon}_w$ for all the vortex particles; $\tilde{\epsilon}_w$ is calculated in VM2D as the square root from the mean value of squared distances to 3 closest vortices (together with above-mentioned markers that are called in VM2D "virtual vortices"). Each vortex particle and each panel of the airfoil's surface line make contributions to the diffusive velocity, as well as to the convective one. However, the diffusive velocity decreases exponentially against the inverse proportional rate of decrease of the convective one. Thus, when calculating the diffusive velocities, it seems to be quite enough to take into account only neighboring vortices and panels, the distance from which is not more than $(5\ldots10)\tilde{\epsilon}_w$, while for convective velocities, contributions of all the particles and panels certainly should be taken into account (exactly or approximately in the framework of any of the fast algorithms).

Note that the values of $\tilde{\epsilon}_w$ are required for viscous friction computation; the corresponding formulae are also derived in the framework of the VVD method. The pressure-based loads are calculated in the VM2D code integrally, as resultant force vector and resultant torque with respect to some specified point (in the VM2D, it is the origin of the local coordinate system, connected with the airfoil shape).

As an example, proving the correctness and a rather high accuracy of viscous effects simulation, let us consider the well-known problem of steady-state flow simulation around the semi-infinite thin plate, which is usually called "the Blasius problem". Some assumptions that are common for the boundary layer theory make it possible to construct its asymptotic analytical solution for the flow velocity field [9,61]. In the numerical simulation, a thin plate of finite length ($L : h = 250 : 1$ aspect ratio) with semicircular tips was considered at the Reynolds number $\text{Re} = V_\infty L / \nu = 10^3$, where $V_\infty$ is incident flow velocity, $\nu$ is its kinematic viscosity. Rather fine discretization of the airfoil and unsteady flow simulation with small time step in the VM2D code lead to the steady-state regime (Figure 3). Hereinafter, red and blue points correspond to vortex particle with positive (counterclock-wise rotation) and negative (clock-wise rotation) circulations, respectively.

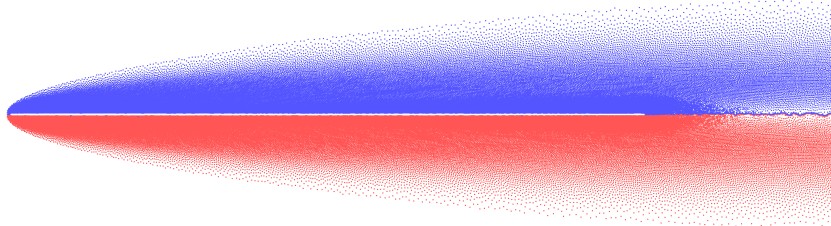

**Figure 3.** Vortex particles positions in steady-state regime for the Blasius flow.

It is seen (Figure 4) that the VPM implementation in the VM2D code, based on the VVD method, allows for very accurate simulation in laminar viscous boundary layers.

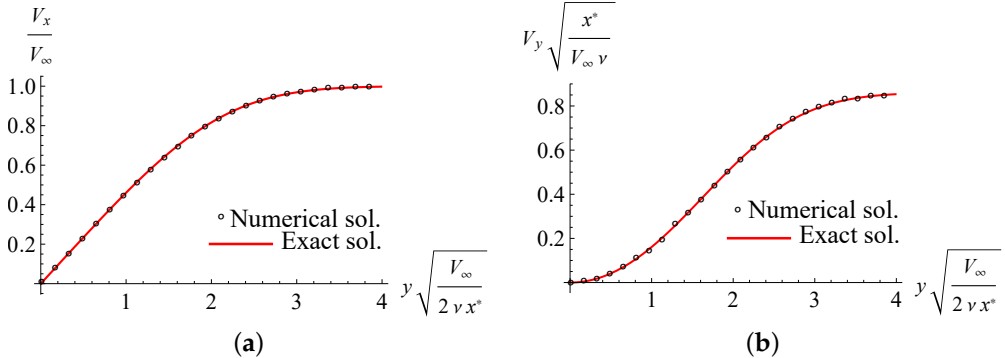

**Figure 4.** Horizontal (**a**) and vertical (**b**) velocity profiles for the Blasius flow in the cross-section distant at one-forth from the leading tip (black circles) and the analytical solution (red solid line).

### 3.3. Vortex Wake Restructuring

Three following operations are also performed on vortex particles; the first and second ones allow for reducing the number of vortex particles in practical computations.

1.　Vortex particles that are located rather far from the airfoil can be excluded from the simulation since they do not practically influence the flow in the near-body region and hydrodynamic loads acting on the airfoil;

2.　Vortex particles that are closer to each other than the preliminary chosen distance $\epsilon_{col}$ are merged ("collapsed") into one summary particle, but only in case the circulation of the summary particle is smaller than $\Gamma_{max}$. Note that the merging of the vortex particles with circulations of different signs is preferable to merging of particles of the same sign; so, the merging subroutine is executed twice in the VM2D code: at first for vortex pairs of opposite signs (without $\Gamma_{max}$ control and for merging distance $\alpha_\epsilon \epsilon_{col}$, where factor $\alpha_\epsilon$ is chosen equal to 2) and then for vortex pairs of the same sign;

3.　Vortex particles penetration control is required since some of the vortex particles intersect the airfoil's surface line. Such vortex particles are excluded from the simulation procedure, while their circulations and intersection points are stored in order to "compensate" the loss of vorticity at the next time step and to take into account correctly the influence of additionally generated circulation (at the next time step) on hydrodynamic loads.

### 3.4. Vorticity Generation on the Airfoil's Surface Line

The mathematical model that lies in the basis of the vorticity generation procedure, wich is used in the VM2D code, differs significantly from the most common approach.

Let us firstly introduce an orthonormal frame for the airfoil's surface line: the normal unit vector $n(r)$ is always directed to the flow domain; $\tau(r)$ is a unit tangent vector, chosen such as $n(r) \times \tau(r) = k$. At corner points and cusp points, there are two frames: limited ones from both sides of such points.

#### 3.4.1. The *N*-Model

In all known modifications of vortex methods, the boundary integral equation, being solved for determination of generated vorticity (that can be represented as concentrated vortices or as distributed vortex sheet), can be obtained as the projection of the BIE (6) onto normal unit vector $n(r)$. The resulting BIE is a singular equation of the first kind with the Hilbert-type kernel $P_n(r, \xi)$

$$\frac{1}{2\pi} \oint_K \underbrace{-\frac{(r - \xi) \cdot \tau(\xi)}{|r - \xi|^2}}_{P_n(r,\xi)} \gamma(\xi) dl_\xi = f_n(r), \quad r \in K, \tag{10}$$

and special quadrature formulae (similar to central points quadratures, but now having lower than the first order of accuracy [40]) should be applied for numerical calculation of its principal values in the Cauchy sense. Here, the right hand side is the projection of (7) onto normal vector: $f_n(r) = n(r) \cdot f(r)$.

Note that the Equation (10) can be derived from the potential theory; the free vortex sheet intensity corresponds to the surface gradient of the double layer potential density:

$$\gamma(r) = n(r) \times \operatorname{Grad} g(r).$$

The Equation (10) always has an infinite set of solutions, so the additional condition (8) should be used to pick out the unique one.

3.4.2. The *T*-Model

The other possible approach is suggested in [31,62] and mentioned at least in [2,40]. It is based on the BIE (6) projection onto tangent vector $\tau(r)$, which leads to the Fredholm-type integral equation of the second kind

$$\frac{1}{2\pi} \oint_K \underbrace{\frac{(r - \xi) \cdot n(\xi)}{|r - \xi|^2}}_{P_\tau(r, \xi)} \gamma(\xi) dl_\xi - \frac{\gamma(r)}{2} = f_\tau(r), \quad r \in K, \tag{11}$$

where $f_\tau(r) = \tau(r) \cdot f(r)$. It is easy to prove that the kernel $P_\tau$ is uniformly bounded in case of smooth airfoil since

$$\lim_{|r - \xi| \to 0} |P_\tau(r, \xi)| = \frac{\kappa(r)}{2},$$

where $\kappa(r)$ is the curvature of the airfoil. In the case of an airfoil with corner points or cusp points the kernel $P_\tau$ is unbounded but absolutely integrable over the arbitrary neighborhood of such point, that allows for developing much more accurate numerical schemes in comparison to the *N*-model.

It is noted in [31] that when implementing the *T*-model, i.e., at solving Equation (11), there is no need to use any additional conditions in opposite to the *N*-model with Equation (10), but this statement is not fully correct. The solution uniqueness for Equation (11) depends on the type of simulated flow:

- For external flows, when the airfoil counterclockwise traversal coincides with tangent vector $\tau(r)$ direction, the unique solution is picked out by condition (8), as earlier;
- For internal flows, when the airfoil clockwise traversal coincides with tangent vector $\tau(r)$ direction, Equation (11) has the unique solution, which satisfies the condition (8), so its consideration together with Equation (11) is unnecessary but remains correct.

However, in [31], only internal flows are considered, so in all those cases there were no problems with solution uniqueness.

Note that the resulting linear system that arises in *T*-schemes can be solved straightforwardly by using Gaussian elimination, or iteratively [63].

3.4.3. BIE Numerical Solution with the *T*-Schemes

Despite the fact that the initial BIE (6) is a vector equation, while Equations (10) and (11) are scalar, all these equations are equivalent.

The numerical schemes, developed for numerical solution of Equation (10) together with (8), are called "*N*-schemes", the schemes for Equation (11) with (8)—"*T*-schemes".

The well-known numerical scheme of discrete vortex method [40], together with some other schemes, used in practice, e.g., described in [2,20], belongs to the *N*-schemes. The most efficient way to the *T*-schemes development is the Galerkin approach implementation, it is mentioned in [31,62]; however, it is impossible to find detailed descriptions of such schemes somewhere. The authors of the present paper have developed such schemes in [52–55] for piecewise-constant and piecewise-linear solution representation over the panels; these schemes provide the first and the second order of accuracy, respectively, in rather "strong" $L_1$ norm for test problems with known exact solutions [64]. Some other variants of the *T*-schemes, including cost-efficient schemes and the schemes of the third order of accuracy for the airfoil, discretized by curvilinear panels, are also developed [65–67]. In the VM2D code the *T*-schemes for rectilinear panels called $\mathcal{T}^0$ and $\mathcal{T}^1$ [55] with piecewise-constant and piecewise-linear solution representation, respectively, are implemented.

It is clear that the higher-order numerical schemes, in general, allow for accuracy improvement. In particular, numerical experiments show that the usage of different schemes influences significantly the hydrodynamic loads acting on the structures in the flow. As an example, let us consider a common test problem of flow simulation around an impulsively started circular cylinder for the Reynolds number Re = 200. The above-

mentioned *N*- and *T*-schemes provide nearly the same results for the initial phase of the vortex wake development while it remains symmetric, Figure 5.

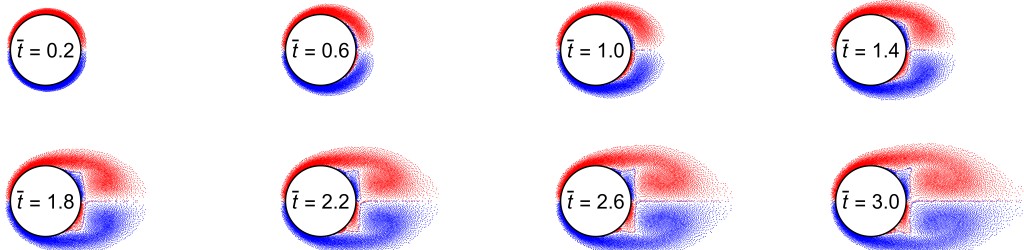

**Figure 5.** Symmetric vortex wake development after the impulsively started cylinder up to dimensionless time $\bar{t} = tV_\infty/D = 3.0$; Re = 200.

The instability of such vortex wake, which consists of two symmetrical vortices after the cylinder, upgrowths, and finally, at steady-state regime, one observes a von Karman vortex street (Figure 6). The time moment at which instability starts to develop in the numerical simulation is different for the considered schemes: the *T*-schemes allow for symmetrical regime simulation slightly longer in comparison with the *N*-scheme.

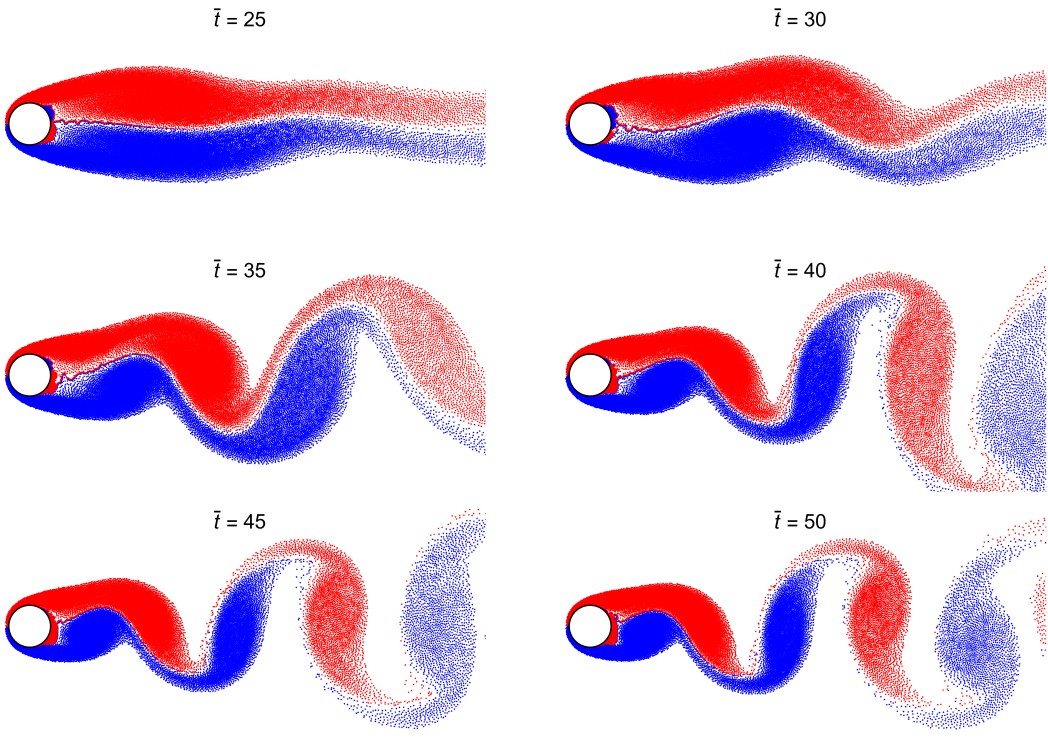

**Figure 6.** Instability development for vortex wake after the cylinder up to dimensionless time $\bar{t} = tV_\infty/D = 50.0$; Re = 200; scheme $\mathcal{T}^0$ is used.

However, the plots of hydrodynamic forces are quite different for the *N*-scheme and *T*-schemes (Figure 7). For identical parameters of the simulation, the forces in the *N*-schemes are oscillating with rather high amplitude; so, it is necessary to implement some filtration technique. At the same time, the *T*-schemes provide much less amplitude of oscillations.

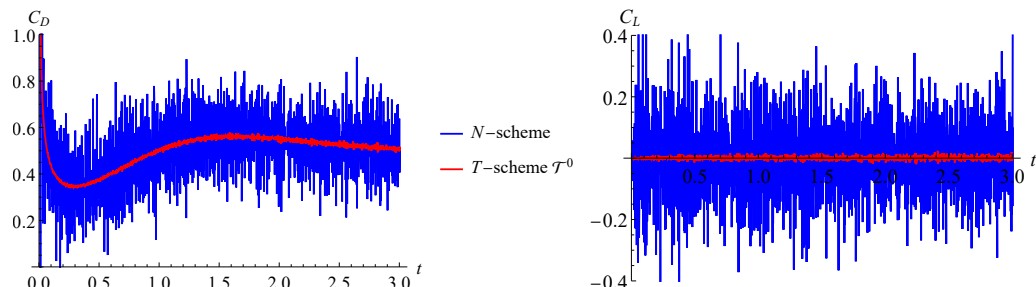

**Figure 7.** Calculated hydrodynamic load oscillations for flow simulation around impulsively starting cylinder: the *N*-scheme and scheme $\mathcal{T}^0$; Re = 200.

The $\mathcal{T}^1$ scheme provides just slightly less level of oscillations for the considered problem; however, if the same simulation is performed with twice smaller time step and twice increased number of panels on the airfoil's surface line, the following result can be obtained: force oscillations for the $\mathcal{T}^1$ scheme become much smaller, than for the $\mathcal{T}^0$ scheme, Figure 8. Thus, it seems that at least for the scheme $\mathcal{T}^1$, there is no need in applying any filters to calculated forces. This feature can be essential when solving FSI-problems.

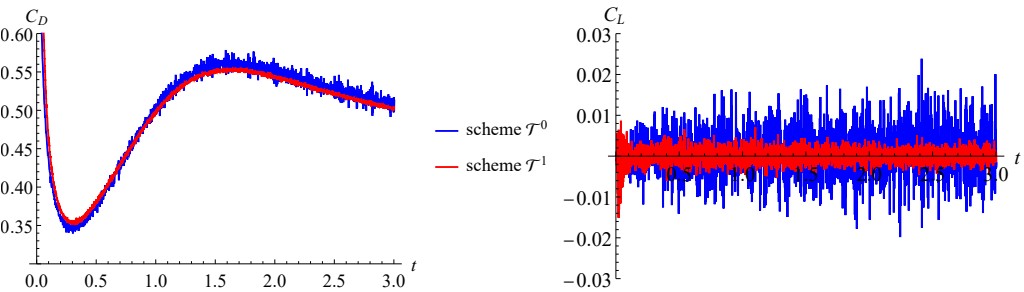

**Figure 8.** Hydrodynamic load oscillations for flow simulation around impulsively starting cylinder: the schemes $\mathcal{T}^0$ and $\mathcal{T}^1$; Re = 200.

Let us also consider a similar problem of flow simulation around impulsively started circular cylinder at Reynolds number Re = 3000. At the initial time moment, the value of the drag coefficient turns out to be very large (Figure 9), then it decreases and starts to increase at approximately $\bar{t} = 0.3$ when vortices start to form. Note that an interesting effect arises in this problem: for a time period between $\bar{t} = 1$ and $\bar{t} = 2$, the value of the drag coefficient stops increasing and then starts to increase again. This effect can be correctly simulated only with a detailed discretization of the flow near the airfoil.

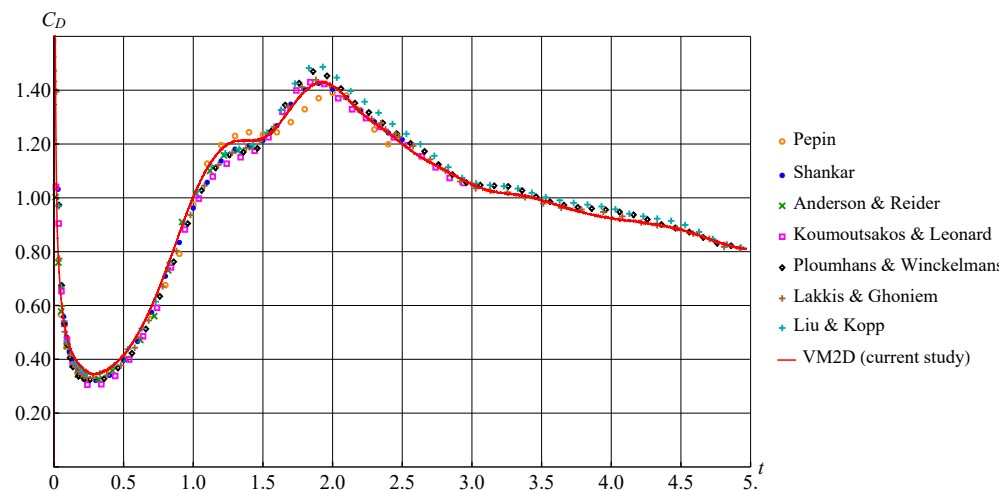

**Figure 9.** Unsteady drag force coefficient for the impulsively started circular cylinder at Re = 3000.

Figure 9 shows how the drag coefficient is changed in time for $0 < \bar{t} < 5$: a comparison of the plot obtained using VM2D with the results of Pepin [68], Shankar [69], Anderson & Reider [70], Koumoutsakos & Leonard [71], Ploumhans & Winckelmans [72], Lakkis & Ghoneim [73] and Liu & Kopp [74] is shown. The flow simulation in VM2D is performed with the following parameters: $N = 3\,200$ panels on the airfoil's surface line, $\Gamma_{max} = 10^{-5}$, $dt = 2.5 \cdot 10^{-4}$. All the parameters hereinafter are dimensionless; spatial scale is the airfoil's diameter $D$; time scale is the diameter divided by incident flow velocity $D/V_\infty$.

It can be seen that the result obtained in VM2D, marked with a red line, is in good agreement with the results of other authors.

## 4. The VM2D Code

In this section, let us describe the developed VM2D code that implements the above described models and approaches.

### 4.1. General Description of the VM2D Code

The code is written in C++ language and has a modular structure. It is a cross-platform software and can be compiled both under Windows and Linux by using MSVC, GCC, Intel C++ Compiler or Clang compilers. External well-known open-source library Eigen is used in VM2D for linear algebra operations. The following parallel technologies are used:

- OpenMP for shared memory systems;
- MPI for distributed memory cluster systems (MPI implementation consistent with the compiler, is required even for computations on shared-memory systems);
- NVidia CUDA technology can be optionally used for graphic accelerators.

The source code is freely available, http://github.com/vortexmethods/VM2D (accessed on 22 February 2023) and it is licensed under free "copyleft" GNU General Public License (GPLv3). After downloading it, CMake software execution is required in order to prepare the necessary makefile or build the project for such environments as MS Visual Studio or Qt Creator as well as for some others.

The functionality of VM2D is mainly aimed at computing unsteady loads acting on the airfoils, as well as solving of the coupled hydroelastic problems when the airfoil (or system of airfoils) moves under the hydrodynamic loads. Problems connected with external flow simulation are more natural for this code, as well as for VPM in general. However, internal flows also can be considered in VM2D: in order to simulate internal flow, for example, inside a tube, Figure 10, the possibility of the point sources introducing is implemented, where positions and intensities of the sources are specified by a user (to provide the divergence-free velocity field in internal flow, the total sum of intensities of all the sources should be equal to zero; sources with negative intensities are treated as sinks). Fixed point sources/sinks also can be introduced into external flow, if it is necessary for some purposes.

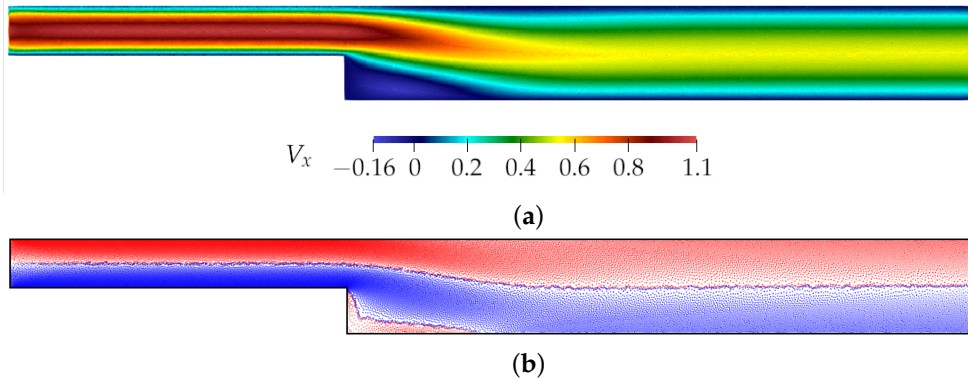

**Figure 10.** Horizontal component of the flow velocity (**a**) and vortex particles positions (**b**) in steady-state regime in the channel with backward-facing step; sources and sinks are placed along the left ("inlet") and right ("outlet") cross-sections, respectively.

It is also possible to consider the problems connected with vortex structures evolution in the flow in the presence or without walls.

### 4.2. The Structure of the `VM2D` Code

The `VM2D` code consists of three libraries called `VM2D`, `VMcuda` and `VMlib`. The `VMlib` library contains descriptions of auxiliary (not directly related to the vortex particle methods) data structures and implementations of some general algorithms, as well as descriptions of universal (abstract) data structures that can be used also for 3D flow simulations according to the closed vortex loops method [75,76]. Names of the last ones are highlighted with the suffix "Gen".

Main classes defined in the `VMlib` library and `VM2D` core are shown schematically in Figure 11 and described below.

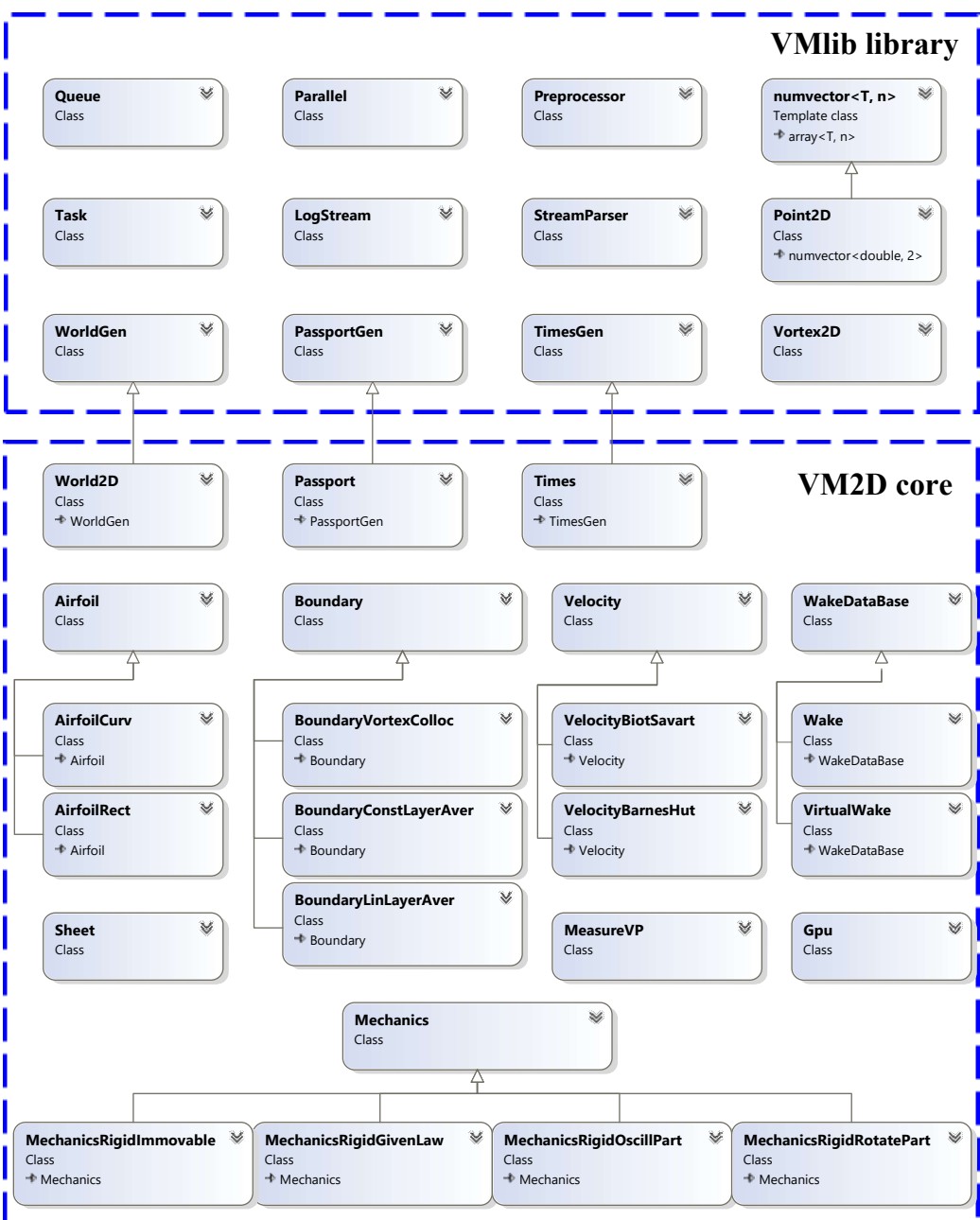

**Figure 11.** Class diagram for `VMlib` library and `VM2D` core.

The `VMcuda` library defines functions that are responsible for calculations performed on GPUs by using the Nvidia CUDA technology. Data structures and classes from the `VM2D` core library represent descriptions of specific objects (airfoil, vortex wake, vortex sheet, etc.) and algorithms used in vortex methods (solving of the boundary integral equation, velocities computation of vortex particles, etc.).

The `VMlib` Library

The `VMlib` library contains classes that describe the computational pipeline: a queue of tasks to be solved, their distribution among MPI-processes, etc. In addition, this library contains auxiliary data structures, which are then used in the core `VM2D` library (e.g., such structures as a 2D vortex particle, geometrical vector, tensor, etc.), as well as abstract parent classes, which are later inherited by a specific implementation for 2D problems.

Classes defined in `VMlib` that provide the computational pipeline are listed below.

`Queue` stores the list of problems to be solved, organizes its solution in MPI parallel mode according to the number of required and available processors, it provides the "external" level of MPI-parallelization: different problems can be solved simultaneously on available cluster nodes;

`Task` stores the state (in the `queue`) of the particular problem and its full description (called hereinafter "passport");

`Parallel` stores the properties of the MPI-communicator created for the particular problem and provides the "internal" level of MPI-parallelization according to which the particular problem is solved in parallel mode on several cluster nodes;

`Preprocessor` is the tool for input file preprocessing; the result is used as input data for `StreamParser`;

`StreamParser` contains the set of tools for input files (after being preprocessed) parsing; it is used for reading all the parameters and initial data, stored in text files;

`LogStream` provides interfaces for the necessary information output (including in parallel mode); note that it is more useful for debugging rather than for typical computations;

`defs` defines the namespace that contains default values for some parameters, the necessary mathematical functions, etc.

The following data structures together with the necessary operations on them have also defined in the `VMlib` library:

`numvector` is a template class for geometric vector that inherits standard wrapper class `std::array<type, n>` and defines the most common operations on vectors (including "&" for scalar product, "^" for vector product, "|" for outer product); `numvector`'s inheritors `nummatrix` and `numtensorX` define fixed size matrix and higher rank tensors, respectively, together with the necessary operations;

`Point2D` inherits `numvector<double, 2>` and has the necessary MPI-descriptor;

`Vortex2D` stores properties of a vortex particle (its position and circulation) and has the MPI-descriptor.

Three abstract classes are introduced:

`WorldGen`—the "sandbox" for each problem of flow simulation being solved;

`PassportGen`—full definition of the particular problem of flow simulation;

`TimesGen`—structures for time statistics assembling and tools for storing to `timestat` file.

### 4.3. The Core Library `VM2D`

The `VM2D` library implements the algorithms for 2D flows simulation and FSI problems solving by using the Vortex Particle Method, namely the Viscous Vortex Domains (VVD) method. Although the general structure of the algorithm is the same as for VVD method, described in the Section 3, it is obvious that some parts of the algorithm can be implemented in different ways (for example, the subroutine for solving the boundary integral equation as well as for velocities computation of vortex particles). To provide the possibility of using

various numerical algorithms, abstract classes and their specific implementations have been developed. The classes defined in `VM2D` are listed in Table 2.

**Table 2.** Classes defined in `VM2D`.

| Class Name | Brief Description |
| --- | --- |
| `World2D` | inherits the `WorldGen` class and describes all the properties and current state of the particular problem from the queue; the instance of this class is the "sandbox" for numerical simulation in 2D problems |
| `Passport` | inherits the `PassportGen` class and stores full definition of the particular 2D problem (its "passport") |
| `Airfoil` | abstract class which describes the geometry of the airfoil |
| `Sheet` | determines attached and free vortex sheets and attached source sheets which are placed on airfoils' surface lines |
| `Boundary` | abstract class that determines the numerical scheme being used for integral equation solution with respect to unknown free vortex sheet intensity |
| `WakeDatabase` | defines the structure for the database, containing the properties of a set of vortex particles (as well as point sources if they are considered); this class is the parent class for the following two classes: |
| | `Wake` that describes the vortex wake in the flow domain; |
| | `VirtualWake` describes vortex particle markers introduced to provide improved transferring of vorticity generated in thin vortex sheet on the airfoil's surface line into the flow domain. |
| `Velocity` | abstract class that determines the numerical scheme for velocities computation in the flow domain |
| `MeasureVP2D` | contains the subroutines and tools for velocity and pressure fields reconstruction in preliminarily specified points in the flow domain |
| `Mechanics` | abstract class that determines the hydroelastic problem and the coupling scheme for its numerical solution |
| `Times` | inherits the `TimesGen` class and adds it with the necessary members for time statistics assembling in 2D problems |
| `Gpu` | provides the possibility to perform calculations on GPU by using the Nvidia CUDA technology |

Four main abstract classes in `VM2D`, namely `Airfoil`, `Boundary`, `Velocity` and `Mechanics` are defined, whose implementations correspond to different modifications of VPM algorithm; some of them are briefly described at the beginning of this paper.

The inheriting classes have names, which consist of the name of the parent class and some additional words that specify the particular implemented method. List of the most important implementations of the abstract classes is given in Table 3. Those implementations that are not fully implemented at that moment are marked by an asterisk; there is a groundwork for their implementation, as well as for a wide range of other methods and approaches.

The `VMcuda` Library

This library contains several subroutines, which are structurally identical to ones, introduced in the `VM2D` library, but are executed on GPU, and also the necessary functions that provide data exchange between the host computer (CPU) and the device (GPU). The implementations of "computational" kernels are in most cases quite different from ones implemented on the CPU due to the specific architecture of the GPUs.

The most time-consuming functions are implemented now for GPUs:

- Convective velocities computation at a set of points induced by vortex particles in the vortex wake as well as by free and attached vortex sheets and attached source sheet;

- Diffusive velocities calculation for vortex particles in the vortex wake and "virtual vortices"—vortex particles introduced to model the vorticity transfer from the free vortex sheet to the flow domain;
- The right-hand side computation for the linear system that arises after discretization of the boundary integral equation on the airfoils' contour lines, which are different depending on the applied numerical scheme;
- Vortex pairs recognizing placed at a rather small distance in the vortex wake for their merging in the framework of vortex wake restructuring algorithm.

**Table 3.** Abstract classes implementations.

| Class Name | Brief Description |
| --- | --- |
| **For class `Airfoil`** | |
| `Rect` | airfoil which surface line is approximated by rectilinear panels |
| `Curv`* | airfoil which surface line is approximated by curvilinear panels |
| **For class `Boundary`** | |
| `VortexColloc` | vortex sheet representation with separate vortex particles, $\mathcal{N}$-scheme, corresponds to the Method of Discrete Vortices for singular integral Equation (10) solution |
| `ConstLayerAver` | piecewise-constant vortex sheet intensity approximation, $\mathcal{T}^0$ numerical scheme for Fredholm-type integral Equation (11) solution |
| `LinLayerAver` | discontinuous piecewise-linear vortex sheet intensity approximation, $\mathcal{T}^1$ numerical scheme for Fredholm-type integral Equation (11) solution |
| **For class `Velocity`** | |
| `BiotSavart` | direct velocity computation by using the Biot–Savart law; $\mathcal{O}(N^2)$ complexity |
| `BarnesHut`* | fast Barnes–Hut/multipole method based on the hierarchical tree construction and traversal; $\mathcal{O}(N \log N)$ numerical complexity |
| **For class `Mechanics`** | |
| `RigidImmovable` | flow simulation around rigid immovable airfoil |
| `RigidGivenLaw` | flow simulation around rigid airfoil at its arbitrary prescribed motion according to the given law |
| `RigidOscillPart` | FSI problem for a rigid airfoil with elastic constraints moving perpendicular to the flow |
| `RigidRotatePart` | flow simulation for a rigid airfoil that rotates under the action of hydrodynamic forces |

The other operations are much less time-consuming, so they are performed on CPUs; however, their transfer to GPU can allow for some additional speedup.

Special data structures are introduced for global memory of GPUs that provide optimized performance, especially for computations in problems with several airfoils in the flow, and also for CPU/GPU data transfer. In most parts of computational subroutines, shared memory is actively used, which allows for a significant increase in performance.

Note that CUDA-implementation is now available for the direct algorithm and not for fast methods. It seems that due to this reason, the implemented possibility of working with the pinned memory (the necessary memory allocator for the host machine is implemented for `std::vector` container) leads just to a very small improvement in performance, and the same for asynchronous concurrent execution for CUDA-kernels, which makes it possible to perform computations simultaneously with data transfer between device and host. Certainly, in the future, fast methods will be also implemented for CUDA (e.g., for the vortex particles velocities computation the algorithm can be developed as a generalization with some modifications of the well-known implementation of the classical Barnes–Hut method, proposed in [77]), so the described features can become more significant.

As for the CPU, all computational subroutines for GPU are implemented for double precision, which decreases significantly the performance of GPUs, especially for `GeForce` and `Quadro` families. Numerical experiments show, however, that some time-consuming operations in Vortex particle methods can be performed with single precision without notable loss of accuracy for the whole simulation. Nevertheless, in the current version of the code, the possibility of computations with single precision is not implemented.

### 4.4. Computational Pipeline

The flowchart of the algorithm is shown in Figure 12. Only time-consuming operations are considered, which are performed at every time step for each task being solved.

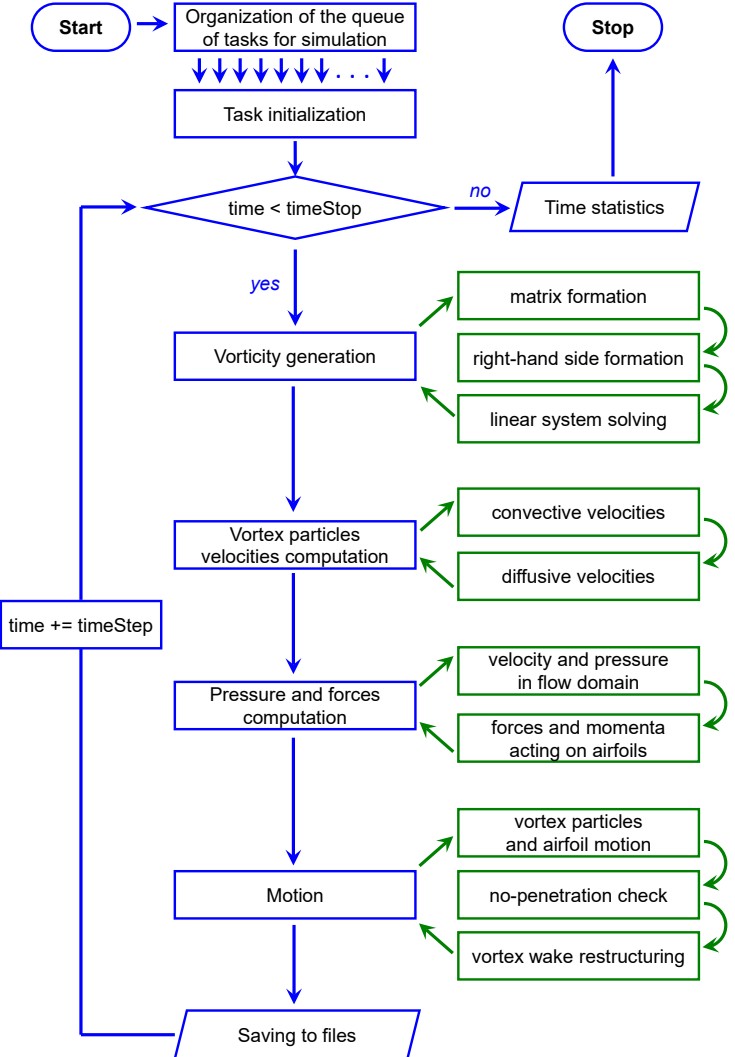

**Figure 12.** Flowchart of the algorithm.

### 4.5. Problems Description in `VM2D`

It is possible to use `VM2D` for the solution of one particular problem as well as for the solution of a set of similar (or not similar) problems. Each problem is denoted with some label (text string without spaces), and a separate subdirectory should be created in the working directory with the name coinciding with the problem's label. Then, all the problems should be listed in the `problems` file, which is placed in the working directory where, if necessary, some parameters can be specified, that will be "passed" to the corresponding problems. The typical structure of the `problems` file is shown in Figure 13.

```
/*-------------------------------*- VM2D -*----------------*--------------*\
| ##   ## ##   ##  ####  #####   |                        | | Version 1.11  |
| ##   ## ### ### ##  ## ## ##   | VM2D: Vortex Method    | 2022/08/07     |
| ##   ## ## # ##    ## ## ##    | for 2D Flow Simulation *----------------*
|  ####  ##   ##   ##   ## ##    | Open Source Code       |                |
|   ##   ##   ## ###### #####    | https://www.github.com/vortexmethods/VM2D |
|                                |                        |                |
| Copyright (C) 2017-2022 Ilia Marchevsky, Kseniia Sokol, Evgeniya Ryatina  |
*--------------------------------------------------------------------------*
| File name: problems                                                       |
| Info: Problems to be solved by using VM2D                                 |
\*-------------------------------------------------------------------------*/
problems = {
wing00deg(np = 1, angle =  0, tau = 1.5e-2),
wing05deg(np = 2, angle =  5, tau = 1.5e-2),
wing10deg(np = 2, angle = 10, tau = 2.0e-2)
};
```

**Figure 13.** The typical structure of the `problems` file.

In the simplest case, it is enough to specify just empty brackets after the problem label (which also can be omitted), but there are two parameters (`pspfile` and `np`) that are always necessary. The `pspfile` defines the name of the passport file with a description of the problem in its subdirectory; the np defines the number of MPI-processes that should be runin parallel for the corresponding program. These parameters can be specified either explicitly, as for `np` in the example in Figure 13, or implicitly, since the default values are defined: `pspfile = passport` and `np = 1`. Note that for every multicore processor, the OpenMP technology is used for parallelization of the algorithm in shared-memory mode.

All other parameters in parentheses are definitions of arbitrary variables of arbitrary type (integer, double, boolean, string, as well as a list), which the user can later use inside "passport" files for their unification and for notational convenience and, as the result, for automatization of similar problems solution.

In the above-considered example, where user has a purpose to solve 3 different problems of flow simulation around wing airfoil (as it follows from the labels of the problems) at different angles of incidence, `passport` file can be the same for all the problems and it can have the structure shown in Figure 14.

All the input files including airfoil geometries and vortex wakes, should be written as dictionaries by using "quasi-C++" syntax: double slash "//" means inline comment, as well as "/* ... */" for multiline comment; line breaks mean the same as spaces; semicolon ";" is the separator between different lines in file; comma "," is the separator in lists and also the separator between parameters in parentheses; spaces and tabs are ignored. The register of parameter names is not sensible.

The sense of the parameters in the passport should be clear, more or less, from their names and short comments in the example in Figure 14.

```
/*-------------------------------*- VM2D -*-----------------*---------------*\
| ##  ## ##   ## #### #####  |                      | Version 1.11   |
| ##  ## ### ### ## ## ## ## | VM2D: Vortex Method  | 2022/08/07     |
| ##  ## ## # ##    ## ## ## | for 2D Flow Simulation  *---------------*
|  #### ##   ##  ##  ## ## | Open Source Code     |
|   ##   ##   ## ##### ##### | https://www.github.com/vortexmethods/VM2D |
|                            |                      |
| Copyright (C) 2017-2022 Ilia Marchevsky, Kseniia Sokol, Evgeniya Ryatina   |
*---------------------------------------------------------------------------*
| File name: passport                                                       |
| Info: Parameters of the problem to be solved                              |
\*-------------------------------------------------------------------------*/

//Physical Properties
rho = 1.0;          // flow density
vInf = {1.0, 0.0};  // incident flow velocity
//vRef = 1.0;       // reference velocity magnitude, isn't used here
nu = 0.001;         // kinematic viscosity coefficient

//Time Discretization Properties
timeStart = 0.0;    // *physical time at which the simulation starts
timeStop = 50.0;    //  physical time at which the simulation stops
dt = $tau;          //  the specific value is set in a batch file, e.g., 1.5e-2
accelVel = Impulse; // *RampLin(T) and RampCos(T) modes are supported

saveVtx = ascii(100); // *frequency (in steps) of vortices storing
saveVP = binary(100); // *and velocity & pressure; text/binary VTK format
nameLength = 5;       // *number of digits in file names

//Wake Discretization Properties
eps = 0.0015;       //  vortex core smoothing radius (should be small)
epscol = 0.0040;    //  vortex particles merging distance
distFar = 20.0;     // *the distance of the vortex wake cropping
delta = 5e-6;       // *distance from airfoil to the generating vortices
vortexPerPanel = 1; // *minimal number of vortices generating over each panel
maxGamma = 1.0e-4;  // *maximal vortex particle strength

//Numerical Schemes
linearSystemSolver = linearSystemGauss;  // *fast methods are under constr.
velocityComputation = velocityBiotSavart;// *Barnes-Hut method is supported
panelsType = panelsRectilinear;          // *curvilin. panels are under constr.
boundaryConditionSatisfaction = boundaryLinearLayerAverage; // *T^1 scheme
// T^0 scheme and N scheme are also available

//Files and Parameters

airfoilsDir = "../settings/airfoils/"; // path to discretized airfoils
wakesDir    = "../settings/wakes/";    // path to vortex wakes

airfoil = {
"naca0012"(                                         // file name
basePoint = {1.0, 0.0}, scale = 1.0,          // geometry
inverse = false,                              // external flow
angle = $angle,                               // AoA
mechanicalSystem = mech0) }; // mechanics

fileWake = { };   // previously stored vortex wake can be loaded
fileSource = { }; // positions and intensities of point sources/sinks
```

**Figure 14.** The typical structure of the passport file.

Not all of them should be specified explicitly since default values for some of them are defined. The default values, having the lowest priority, are introduced directly in the source code, such parameters marked with the asterisk in the comment in the above-given example in Figure 14; their default values are the following:

| | | | |
|---|---|---|---|
| `timeStart` | `= 0.0` | `distFar` | `= 10.0` |
| `accelVel` | `= RampLin(1.0)` | `delta` | `= 1.0 × 10`$^{-5}$ |
| `saveVtx` | `= ascii(100)` | `vortexPerPanel` | `= 1` |
| `saveVP` | `= ascii(0)` | `maxGamma` | `= 0.0` |
| `nameLength` | `= 5` | | |

The defaults with stronger priority for these and other parameters can be specified by the user in the `defaults` file that should be placed in the working directory. Verbal expressions of some options (such as `velocityBiotSavart`, `boundaryConstantLayerAverage` and others) can be defined in file `switchers`, also placed in the working directory.

Let us give here short descriptions of some parameters which sense can be not trivial:

`vRef` is reference velocity magnitude; it is required in problems without incident flow as a scale for dimensionless parameters, otherwise, it is equal to the magnitude of the incident flow velocity;

`accelVel` defines the way of the influence flow acceleration from zero to `vInf` value; it can take the following values:

- `Impulse` means that the flow starts instantly (impulsively);
- `RampLin(T)` means that the flow is accelerated linearly from zero to `vInf` during *T* seconds;
- `RampCos(T)` means that the flow accelerates according to the cosine law from zero to `vInf` during *T* seconds;

`saveVtx`, `saveVP` zero values mean that the corresponding files should not be saved at all; `maxGamma` zero value means no limitation for $\Gamma_{max}$—maximal value of the vortex particles intensity.

In order to save the velocity and pressure fields to files, it is necessary to list the points for their computation in the file `pointsVP` in the problem's directory.

Files with airfoils geometry are stored in `airfoils` directory; they are text files with very simple format, which is absolutely clear from tutorial examples. For the airfoils, the following parameters should be specified after the file name inside the brackets:

- `basePoint`—point at which the airfoil center should be placed;
- `scale`—scale factor for the airfoil;
- `inverse`—boolean switch for internal flow simulation inside the airfoil;
- `angle`—angle of incidence;
- `mechanicalSystem`—numerical scheme for coupling strategy implementation in coupled FSI problems.

Note that in the example of the passport shown in Figure 14, two parameters are not defined explicitly: the time step `dt` and angle of incidence of the airfoil `angle`. Such templates are marked by symbol "$", which means that their values are equal to user-defined variables, which are defined in the previously shown in Figure 13 file `problems` inside the parentheses after labels of the corresponding problems.

Moreover, in order to solve several similar problems, which differ only by some parameters values, it is suitable to specify `copyPspFile` option in the `problems` file, for example, as it is shown in Figure 15. As a result, all the necessary subdirectories for the listed problems will be created automatically, and the files contained in the specified folder will be copied there (there should be at least `passport` file, and `pointsVP` file, if necessary); obviously, the passport file for such problems should contain templates for some parameters.

```
problems = {
wing00deg(np = 1, angle =  0, tau = 1.5e-2, copyPath="./wingBase"),
wing05deg(np = 2, angle =  5, tau = 1.5e-2, copyPath="./wingBase"),
wing10deg(np = 2, angle = 10, tau = 2.0e-2, copyPath="./wingBase")
};
```

**Figure 15.** The example of the `copyPath` parameter usage.

If the flow around a system of airfoils is simulated, more than one airfoil file should be specified. In this case, the corresponding section of the passport file has the structure of a list, Figure 16:

```
airfoil = {
square_160points(
basePoint = {0.0, 0.0},
angle = 45.0,
scale = 1.0,
mechanicalSystem = mech0
),
circle_200points(
basePoint = {1.2, -0.2},
angle = 0.0,
scale = 0.5,
mechanicalSystem = mech0
)
};
```

**Figure 16.** Code for flow around system of airfoils simulation.

In this example, the interference phenomenon for two immovable airfoils is simulated: small (2 times scaled) circular airfoil placed behind the square airfoil (installed "rhombic" at the angle of incidence 45°) in its vortex wake. All other parameters will be set to default values.

The `mechanicalSystem` parameter can take different values that are labeled for different types of mechanical systems. At the moment, a few different types are implemented and their labels are decrypted in the `mechanics` file (Figure 17), where each label is assigned with the name of the class, which inherits the abstract `mechanics` class (a brief description of these classes is given in Table 3).

```
mech0 = mechanicsRigidImmovable();
mech1 = mechanicsRigidGivenLaw();
mech2 = mechanicsRigidOscillPart(sh={0, $shy}, m=$m);
mech3 = mechanicsRigidRotatePart(J=$J, shw=$shw, Mz=$Mz);
```

**Figure 17.** The `mechanics` file structure.

A user can also pass the specific parameters of the mechanical system in parentheses. For translatory motion with two degrees of freedom, the object `mechanicsRigidOscillPart` takes the following parameters: dimensionless eigenfrequencies `sh` of oscillations in horizontal and vertical directions, and airfoil's mass `m`. For rotational oscillations, the object `mechanicsRigidRotatePart` takes the moment of inertia `J`, dimensionless eigenfrequency `shw` and load torque `Mz`. These values are equal to user-defined variables, which are defined in file `problems`.

As an example of simulation using mechanical system `mechanicsRigidRotatePart`, we consider the flow around a rotating Savonius rotor. The Savonius rotor is an example of one of the simplest types of turbines and it is well investigated both experimentally and numerically [78–80].

A Savonius rotor with two blades is considered with the following parameters: diameter $D = 2$, blade thickness $h = 0.05$, polar inertia moment $J = 10$, incident flow velocity $V_\infty = 1$, Reynolds number $\text{Re} = \frac{DV_\infty}{\nu} = 10^5$. Motion of the Savonius rotor was carried out according to the following scheme: the rotor angular velocity was linearly increased from 0 to 2 during the first 15 s, then, up to 45 s, the rotor rotated under the action of hydrodynamic loads only, from 45 s, the rotor was loaded with a torque $M_z = -0.25$. As a result, after a while the angular velocity of the Savonius rotor stabilized at approximately 1.35. The angular velocity $\omega$ of the Savonius rotor rotating according to the described scheme is shown in Figure 18.

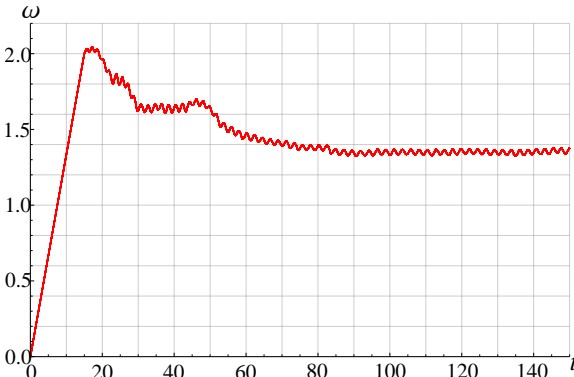

**Figure 18.** Angular velocity of Savonius rotor loaded with a torque $M_z = -0.25$.

The Figure 19 shows the vortex wake around the rotating Savonius rotor at the beginning of the numerical simulation ($\bar{t} = 6$) and developed vortex wake after some simulation time ($\bar{t} = 32$).

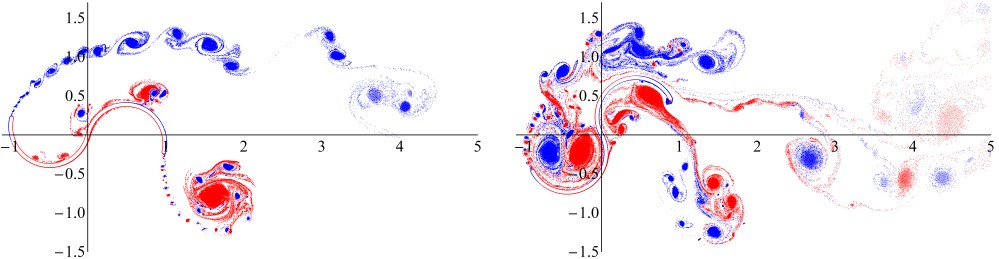

**Figure 19.** Vortex wake at the beginning of simulation (left figure, $\bar{t} = 6$) and developed vortex wake (right figure, $\bar{t} = 32$) around the rotating Savonius rotor ($\text{Re} = 10^5$).

Comparison with OpenFOAM has been performed for a similar problem of rotor autorotation simulation; unsteady lift and drag forces are shown in Figure 20; at $t = 0$, the rotor had zero angular velocity. It is seen that results are in acceptable agreement.

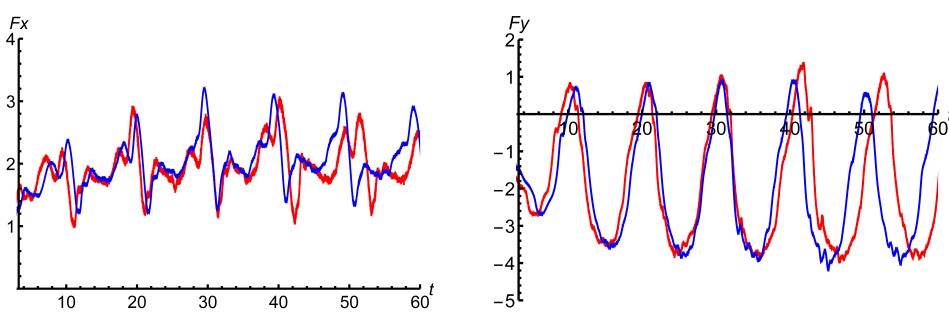

**Figure 20.** Unsteady lift and drag forces for an autorotating rotor; —— VM2D, —— OpenFOAM.



If the user wants to use a previously simulated vorticity distribution, it can be uploaded by specifying the corresponding file name in section `fileWake` of the passport. Files with vortex wakes description should be stored in `wakes` directory.

Some tutorial examples can be found in `tutorials` folder on `github`; in the `run` folder, the examples of files that should be placed in the working directory are also given.

### 4.6. Documentation

Programmer's guide to the `VM2D` code is generated automatically by using `doxygen` tool. It includes full information about all the classes implemented in `VM2D`: description of all the class members and class methods. Relationships between the classes are also shown in graphical mode, as well as execution diagrams of the functions. Html-version of documentation is available at http://vortexmethods.github.io/VM2D/ (accessed on 22 February 2023) and it is being updated automatically via `Travis-CI` service after every modification of source code and its push on `GitHub`.

The user's guide to the `VM2D` code is prepared by using the `Read the Docs` service. This guide describes the process of installing and running of `VM2D`. Its current version is available at https://vm2d.readthedocs.io/ (accessed on 22 February 2023).

### 4.7. Results of Simulation

The results of simulation are saved in files in the problem's directory. Files that contain the description of vortex wake at particular time steps (every `saveVtx` steps) are saved in subdirectory `snapshots` in ascii or binary vtk-format. If `saveVtx=0`, then the files are not saved. Similarly, files containing velocity and pressure fields, are saved to the `velPres` directory. If it is required, the velocity and pressure field files can be processed with the POD technique in order to reduce the amount of data [81,82].

Integral hydrodynamic loads (total force vector and the torque) acting on the airfoils are saved for all time steps in `forces-airfoil-n` files, separately for pressure-based components and viscous friction ones; positions and velocities of the moving airfoils are saved in `position-airfoil-n` files. The mentioned files are saved both in text format as columns and in `csv` format.

It is recommended to use the open-source cross-platform `Paraview` package, which provides a lot of possibilities for visualization of the results of simulation.

Time statistics is being stored in `timestat` file; the program log is shown on screen. It can be redirected to a file by using standard command prompt/shell operator >.

### 4.8. Parallelization of Computations

As mentioned earlier, all time-consuming operations in `VM2D` are parallelized by using the OpenMP and MPI technologies, which allow performing calculations on multi-core/processor systems, including ones with distributed memory. Some most time-consuming subroutines are parallelized by using Nvidia CUDA technology. Its usage in `VM2D` turns out to be rather efficient, since the particles can be processed independently in many computational blocks of the algorithm (usually, the number of particles is sufficiently high in simulations).

Let us estimate the efficiency of parallelization. In the first model problem, the development of the vortex wake after the circular airfoil (Figure 5) was simulated for Re = 200 on the shared-memory computational node with two 18-core Intel Xeon Gold 6254 processors.

The circular airfoil was discretized into 1000 panels, and step execution time was measured for different time steps with different numbers of vortices in the wake. The graphs for achieved speedup is shown in Figure 21 for different number of vortices in the wake. For a rather high number of vortex particles, a close to linear speedup is observed.

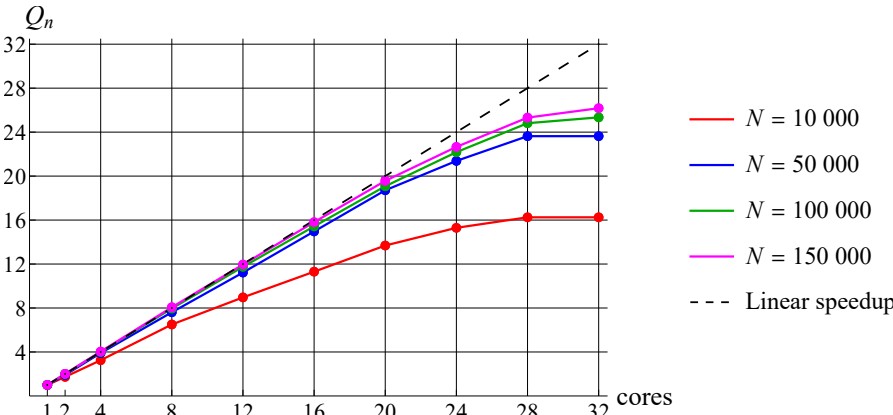

**Figure 21.** Speedup for different number of cores with OpenMP technology for flow simulation with different resolution; *N* is the approximate number of vortex particles in the wake.

The calculations were carried out on a different number of cluster nodes, each equipped with 28 cores (2 × Intel Xeon E5-2690v4). In order to estimate the efficiency of the MPI parallelization, let us consider flow around a circular cylinder for Re = 3000. The circular airfoil was discretized by 2000 panels, the time-step was chosen as $dt = 0.0005$. In this simulation, due to the small value of maximal circulation of vortex elements, $\Gamma_{max} = 0.00002$, the number of vortex elements increases rather quickly and after the first 10 steps the value of 200,000 is already reached.

Speedup for different number of 28-core nodes with the MPI technology for flow simulation with different resolutions was examined; *N* is the average number of vortex particles. To estimate the speedup of the algorithm the execution time of the first 500 time steps is analyzed for different number of cores. Figure 22 shows the speedup of the algorithm for the described problem, where the maximum number of vortex elements was approximately 300,000, as well as the speedup for a similar problem where the number of vortex elements was approximately twice as large. It can be seen that the speedup of problems with 300,000 and 600,000 vortices approximately coincides with the speedup corresponding to Amdahl's law with 0.65 and 0.45% of the sequential code, respectively.

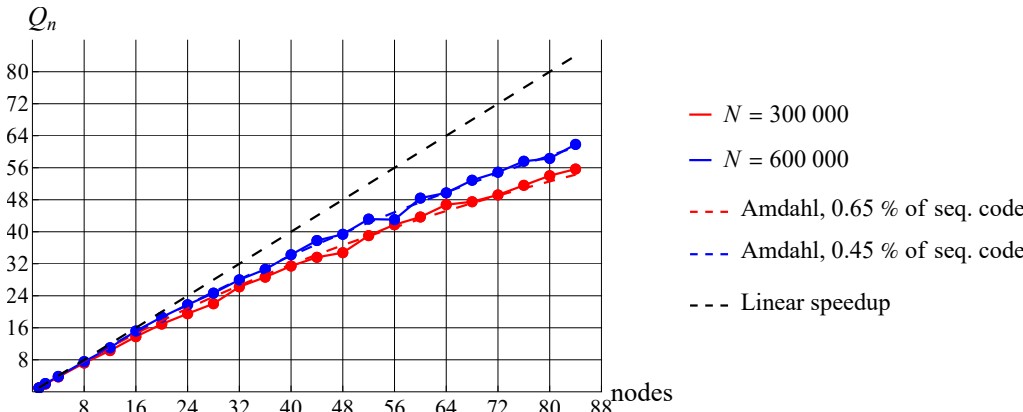

**Figure 22.** Speedup for different number of 28-core nodes with the MPI technology for flow simulation with different resolution; *N* is the average number of vortex particles.

The main time-consuming operations that can be distinguished in the algorithm of VM2D are shown above in Figure 12. Italic type below indicates the name of the column in the `timestat` file, which contains the execution time of the corresponding operation for every time step of simulation.

- *tMatrRhs*—calculating the coefficients of matrix and right-hand side of linear system;
- *tSolve*—solving the linear system (by Gaussian elimination);

- *tConvVelo*—calculating convective velocities of vortices;
- *tDiffVelo*—calculating diffusive velocities of vortices;
- *tForce*—calculating the hydrodynamic forces acting on the airfoils;
- *tVelPres*—calculating the velocity and pressure fields in the specified points;
- *tMove*—calculating new positions of vortex particles;
- *tInside*—detecting the vortices trapped inside the airfoil after movement;
- *tRestr*—vortex wake restructuring (merging closely spaced vortices, removing from the simulation of vortices that are too far from the airfoils, etc.);
- *tSave*—saving data to files.

In the Table 4, execution times for most time-consuming operations are given for simulation on 1 node ($T_1$) and 84 nodes ($T_{84}$). In the second row, fractions of execution time that correspond to mentioned operations are given for simulation on one node. Execution time taken to complete all other operations is shown in the last column labeled *tOthers*. The last row shows the speedup $T_1/T_{84}$ of operations obtained using 84 nodes in comparison to 1 computational node.

**Table 4.** Speedup of `VM2D` algorithm operations using MPI.

|  | *tStep* | *tMatrRhs* | *tConvVelo* | *tDiffVelo* | *tInside* | *tRestr* | *tOthers* |
|---|---|---|---|---|---|---|---|
| fraction, % | 100 | 0.56 | 79.05 | 9.31 | 0.10 | 10.96 | 0.02 |
| $T_1$, sec. | 54.564 | 0.306 | 43.134 | 5.081 | 0.056 | 5.978 | 0.009 |
| $T_{84}$, sec. | 1.014 | 0.019 | 0.640 | 0.114 | 0.014 | 0.186 | 0.041 |
| $T_1/T_{84}$ | 53.8 | 15.9 | 67.4 | 44.6 | 4.2 | 32.1 | 0.2 |

It can be seen from the Table 4 that the most significant speedup is observed for the operations of calculating the convective and diffusion velocities and restructuring of the vortex wake; it should be noted that these operations are also the most time-consuming.

For the same problem, time tests were performed on computer systems with various GPUs in order to estimate the efficiency of parallelization computations in `VM2D` using graphics accelerators. The following GPUs were used for computational experiments: Quadro P2000, GeForce GTX 970, Tesla C2050, Tesla V100, Tesla A100. Table 5 shows the properties of used GPUs and average execution time $T$ of one time step averaged over 700 first steps of simulation.

**Table 5.** Computational time for different GPUs.

| GPU | Quadro P2000 | Geforce GTX 970 | Tesla C2050 | Tesla V100 | Tesla A100 |
|---|---|---|---|---|---|
| Multiproc. | 8 | 13 | 14 | 80 | 108 |
| CUDA-cores | 1024 | 1664 | 448 | 5120 | 6912 |
| GFlops, double | 94 | 123 | 515 | 7450 | 9700 |
| $T$, sec. | 27.1 | 25.7 | 19.5 | 0.97 | 0.85 |

The `VM2D` code provides the possibility of using several graphics accelerators to perform one simulation; the communication between video cards is carried out using the MPI technology. The efficiency of this feature was estimated using tests with two types of GPUs: Tesla C2050 and Tesla A100. Figure 23 shows the speedup of the evaluation time of the algorithm time step when using a different number of Tesla C2050 GPUs compared to one Tesla C2050 GPU (each node of the cluster was equipped with 3 graphics accelerators). Separate lines correspond to different numbers of vortex elements in the wake.

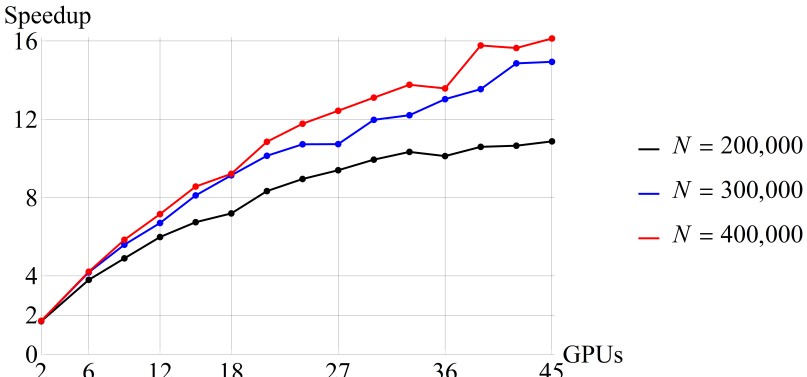

**Figure 23.** Speedup for different number of Tesla C2050 GPUs (horizontal axis); *N* is the average number of vortex particles in the wake.

Figure 24 shows the speedup with Tesla A100 GPUs depending on the number of vortex particles in the wake. Color lines correspond to different number of GPUs.

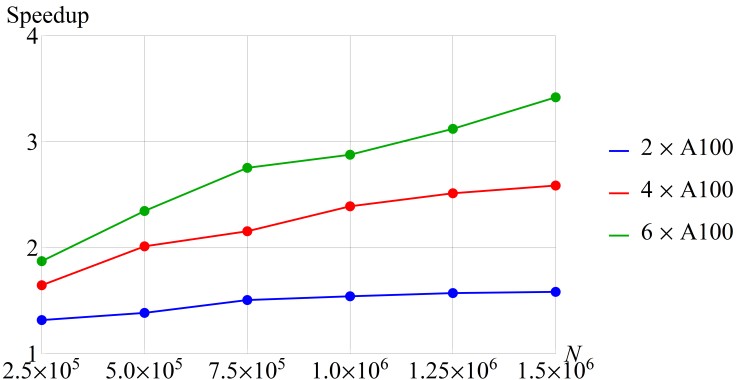

**Figure 24.** Speedup for different number of Tesla A100 GPUs (2, 4, and 6); *N* is the average number of vortex particles in the wake (horizontal axis).

The results of performed numerical experiments have shown that the VM2D algorithm is quite scalable: on a cluster system with 2500 CPU cores, the parallelization efficiency reaches approximately 0.7. Since the vortex method implemented in VM2D belongs to a class of particle-based methods, the algorithm is quite efficiently adapted for graphics accelerators: it has been shown that one Tesla V100 graphics card in terms of performance in VM2D can replace 84 28-core CPU nodes. In addition, it is possible to use several GPUs in one simulation, although the use of several cards is not very efficient, but if time of computations is critical, it provides an additional speedup of calculations.

## 5. Conclusions

The paper describes a code VM2D developed for the simulation of two-dimensional viscous incompressible flows, estimation of unsteady hydrodynamic loads acting on immersed airfoils, as well as for solving coupled FSI problems. Immersed airfoils can move arbitrarily, translationally, and/or rotationally, according to a given law or under the action of hydrodynamic loads. Problems connected with internal flow simulation can also be considered.

The VM2D is open source code; user and developer guides are available. The code is written in C++, has a modular structure, and supports OpenMP, MPI, and Nvidia CUDA parallel computing technologies. The code is based on the Viscous Vortex Domain method, which belongs to the class of fully Lagrangian vortex particle methods, which leads to good scalability of the code, including for graphics accelerators.

In typical simulations, the number of vortex particles in the vortex wake has an order of hundreds of thousands and even millions; parallelization, of course, is the more efficient, the more particles are in the wake. Numerical experiments show that for 600,000 vortex particles, the parallelization efficiency for 2500 CPU cores is about 0.7. Parallel algorithms for graphics accelerators are developed for the most time-consuming operations: computation of the vortex particles velocities, the vortex wake influence on the airfoil accounting and finding pairs of closely-placed particles. As a result, a simulation performed on one Nvidia Tesla V100 GPU takes about the same time as a similar simulation using more than 2000 CPU cores. Multiple GPUs together with MPI usage are also available.

**Author Contributions:** Conceptualization, I.M. and K.S.; formal analysis, I.M., K.S., E.R. and Y.I.; investigation, I.M. and K.S.; methodology, I.M., K.S. and E.R.; software, I.M., K.S. and E.R.; supervision, I.M.; verification, I.M., K.S. and Y.I.; visualization I.M., K.S. and E.R.; writing—original draft, I.M., K.S., E.R. and Y.I.; writing—review and editing, I.M. and K.S. All authors have read and agreed to the published version of the manuscript.

**Funding:** This paper has been supported by the Kazan Federal University Strategic Academic Leadership Program ("PRIORITY-2030").

**Data Availability Statement:** Data is contained within the article. The code and tutorials are freely available on GitHub: http://github.com/vortexmethods/VM2D (accessed on 22 February 2023).

**Conflicts of Interest:** The authors declare no conflict of interest. The funders had no role in the design of the study; in the collection, analyses, or interpretation of data; in the writing of the manuscript, or in the decision to publish the results.

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
