# Peer review of "The VM2D Open Source Code for Two-Dimensional Incompressible Flow Simulation by Using Fully Lagrangian Vortex Particle Methods"

_axioms, doi:10.3390/axioms12030248_

Round 1
Reviewer 1 Report
The authors have reported on a code "VM2D" developed for the simulation of two-dimensional viscous incompressible flows. The code has been proposed to be capable of estimating the unsteady hydrodynamic loads acting on immersed airfoils and solving coupled FSI problems. Problems connected with internal flows can also dealt with using the newly developed code. The paper is reasonably well written and can be considered for publication after a revision.
The authors should address the following in the revised paper:
1. In section 4.2, the structure of the code can be better explained using a flow chart that illustrates the different subroutines and functionality of the code.
2. Support the results shown in figure 11 with some quantitative results that validate the capability of the code.
3. Please add linear speedup curve to figure 12 and 13.
Author Response
The paper is revised; all the things pointed out by the Reviewer are corrected:
- The class diagram and the flowchart are added (fig. 10 and fig. 11).
- For the considered problem of the Savonius rotor autorotation a comparison with OpenFOAM for unsteady drag and lift forces is added (fig. 14).
- Linear speedup curve is added to figures 15 and 16.
Reviewer 2 Report
The manuscript entitled “The VM2D open source code for two-dimensional incompressible flow simulation by using fully Lagrangian vortex particle methods” has been reviewed. This manuscript described the open-source C++ code VM2D for the simulation of two-dimensional viscous incompressible flows and solving fluid-structure interaction problems. In this manuscript, first of all, authors introduced the advantages of VPM method, listed some software using VPM. Then, authors introduced the mathematical formulation of the problem of modeling two-dimensional flows and solving FSI problems. In next section, authors gave a brief history of VPM, a description of their modification-the VVD method, numerical schemes for solving the boundary integral equation. The solutions of “the Blasius problem” and “impulsively started cylinder” are also given in this section. The solution of “the Blasius problem” showed that the VVD method allows for very accurate simulation in laminar viscous boundary layers. The solution of “impulsively started cylinder” showed that different schemes (N-scheme or T-schemes) may lead to different solution of hydrodynamic forces. Finally, authors described the structure of VM2D code, core library of VM2D, input/output files, how to view the document of VM2D and showed the efficiency of parallelization using the CPU and GPU.
Although this manuscript is detailed, the accuracy of the calculation results has been verified, there are still some suggestions for the authors to consider before publishing.
My comments are listed as follows:
1. In the first section, authors introduced some existing software. The authors can set a table to compare the functions and features of VM2D and other software, and emphasize the innovation and advantages of VM2D, which can make the text more concise.
2. What is the meaning of particles ‘color (blue/red) in Figure 3, Figure 4, Figure 5, Figure 9 and Figure 11? Please explain it in text or the title of figure.
3. As a solution of transient problem, the author should indicate the time in each figure in Figure 5. In addition, the authors mentioned that the solutions under different schemes (T-schemes or N-scheme) are slightly different (T-schemes allow for symmetrical regime simulation slightly longer in comparison to the N-scheme). The authors should point out the scheme in which Figure 5 is used.
4. In Figure 6, the authors gave two CD curves. What is the difference between the left one and the right one?
5. Above the Figure 11, the authors mentioned “after some simulation time”. The authors should replace “after some simulation time” with an accurate time.
6. At the beginning of the fourth section, the authors should give the flow chart of VM2D and point out which files or libraries are used in each part. Through this method, readers will have a more intuitive understanding of VM2D.
7. On page 26, authors mentioned time-step dt=0.0005 and Γmax=0.00002. Are dt and Γmax dimensionless? If not, please add the unit. Please the authors check whether other physical quantities in this manuscript also have such problem.
Author Response
The paper is revised; all the things pointed out by the Reviewer are corrected:
- The table is added in introduction with brief comparison of known 2D vortex particle codes (table 1).
- Color means vortex particles with positive and negative circulation; explanation is added before Fig 3.
- Time is indicated in figs. 4 and 5, the scheme used in fig. 5 (T^0) is mentioned.
- It was a typo; "C_L" instead of "C_D" is written on right plots in figs. 6 and 7.
- Simulation time is given for results shown before fig. 13 (former fig.11)
- The class diagram and the flowchart are added (fig. 10 and fig. 11).
- All the parameters are dimensionless, it is mentioned on page 16, before fig. 8.
Reviewer 3 Report
1. Highlight the article innovation point in the abstract. Why should anyone read your article?
2. The Abstract should contain answers to the following questions: What problem was studied and why is it important? What methods were used? What are the important results? What conclusions can be drawn from the results? What is the novelty of the work and where does it go beyond previous efforts in the literature? Please include specific and quantitative results in your Abstract, while ensuring that it is suitable for a broad audience.
3. Brief description of each condition and assumption employed need to be addressed clearly.
4. There are concerns about symbols and equations through manuscript.
5. Quality of the graphs are very low. Keep it for 300dpi and try to smoothen the lines of the graphs.
6. It is helpful to complete the description of how to collect data, data processing scenarios, and interpret the data collection.
7. Revise the introduction such that each paragraph shall present the meaning of a concept/keyword
8. Figures: First, the self-explanatory legend is required. There is not enough info on the drawings. That means, I would like to see more connection between the drawings and the model including boundary conditions. Finally, by looking into the figure it would be good to get some visual representation of the physical process the authors consider.
9. I would suggest to leave in the main text the final version of the model (the one which is subject to numerical solution) and move all the technical details (maybe adding more rigorous derivations) into the supplementary materials.
10. How to reliable on your model.
11. Use vector graphic images, and avoid serif fonts in figures (use sans-serif types).
12. Justify the use of the classical fluid models in light of new experimentally verified models.
Author Response
The paper is revised; the reviewer's comments related to this article are taken into account
Reviewer 4 Report
A useful and detailed paper focusing on the applicability of the Vortex Particle Method. My comments are mainly related to the formatting of the text:
- - Currently the reference list is in alphabetical order, but it is more common to follow the order of the appearance
- - Not all of the abbreviations are explained at the first appearance (e.g FSI)
- - Sometimes the numbering of the equations is missing
- - Please consider to place the code before the results it generated
Author Response
The paper is revised; all the things pointed out by the Reviewer are corrected:
- Reference list is now in the order of the appearance.
- Abbreviations are explained at the first appearance (FSI, LES, BIE, and all other).
- Only those equations are numbered which are referenced in the paper.
- The results which are placed in the paper before code description, are connected with some specific algorithms, implemented in the code; the examples, that show the performance of the whole code are placed below the code description.